# A multifaceted role of progranulin in regulating amyloid-beta dynamics and responses

Huan Du[1], Man Ying Wong[3], Tingting Zhang[1], Mariela Nunez Santos[1], Charlene Hsu[1], Junke Zhang[2], Haiyuan Yu[2], Wenjie Luo[3], Fenghua Hu[1]

**Haploinsufficiency of progranulin (PGRN) is a leading cause of frontotemporal lobar degeneration (FTLD). PGRN polymorphisms are associated with Alzheimer's disease. PGRN is highly expressed in the microglia near Aβ plaques and influences plaque dynamics and microglial activation. However, the detailed mechanisms remain elusive. Here we report that PGRN deficiency reduces human APP and Aβ levels in the young male but not female mice. PGRN-deficient microglia exhibit increased expression of markers associated with microglial activation, including CD68, galectin-3, TREM2, and GPNMB, specifically near Aβ plaques. In addition, PGRN loss leads to up-regulation of lysosome proteins and an increase in the nuclear localization of TFE3, a transcription factor involved in lysosome biogenesis. Cultured PGRN-deficient microglia show enhanced nuclear translocation of TFE3 and inflammation in response to Aβ fibril treatment. Taken together, our data revealed a sex- and age-dependent effect of PGRN on APP metabolism and a role of PGRN in regulating lysosomal activities and inflammation in plaque-associated microglia.**

## Introduction

Progranulin (PGRN), encoded by the *granulin* (*GRN*) gene in humans, has been shown to be a key player in brain aging and neurodegenerative diseases. First, PGRN haploinsufficiency resulting from mutations in the *GRN* gene, is one of the major causes of frontotemporal lobar degeneration (FTLD) with TDP-43 positive inclusions (Baker et al, 2006; Cruts et al, 2006; Gass et al, 2006). Second, complete loss of PGRN in humans is known to cause neuronal ceroid lipofuscinosis (NCL) (Smith et al, 2012; Almeida et al, 2016), a group of lysosomal storage diseases. Third, a recent study identified *GRN* as one of the two main determinants of differential aging in the cerebral cortex with genome-wide significance (Rhinn & Abeliovich, 2017). Fourth, *GRN* is one of the five risk factors for a recently recognized disease entity, limbic-predominant age-related TDP-43 encephalopathy (LATE) (Nelson et al, 2019). Finally, PGRN polymorphisms contribute to the risk of Alzheimer's disease (AD) (Kamalainen et al, 2013; Perry et al, 2013; Sheng et al, 2014; Xu et al, 2017), and serum PGRN levels are inversely proportional to the risk of AD development (Hsiung et al, 2011).

PGRN is an evolutionarily conserved, secreted glycoprotein of 88 kD comprised of 7.5 granulin repeats (Bateman & Bennett, 2009; Nicholson et al, 2012). The full-length precursor can be proteolytically cleaved into several biologically active granulin peptides (granulins A, B, C, D, E, F, G), which have distinct activities in some cases (Tolkatchev et al, 2008; Holler et al, 2017). Multiple recent studies have suggested a critical role of PGRN in the lysosome. Since its genetic link to NCL was discovered (Smith et al, 2012; Almeida et al, 2016), lysosome dysfunction and lipofuscin accumulation have also been reported in PGRN knockout mice (Ahmed et al, 2010; Tanaka et al, 2014; Gotzl et al, 2018). FTLD patients with *GRN* mutation have also been reported to have NCL-related phenotypes (Gotzl et al, 2014; Valdez et al, 2017; Ward et al, 2017). Within the cell, PGRN is localized in the lysosome compartment (Hu et al, 2010). PGRN traffics to the lysosome via direct interactions with the trafficking receptor sortilin or the lysosomal protein prosaposin (Hu et al, 2010; Zhou et al, 2015, 2017c). In the lysosome, PGRN is processed to granulin peptides by cathepsins (Holler et al, 2017; Lee et al, 2017; Zhou et al, 2017b). Whereas the functions of granulin peptides remain to be fully characterized, PGRN and granulin peptides have been shown to regulate lysosome enzyme activities, such as cathepsin D (Beel et al, 2017; Valdez et al, 2017; Zhou et al, 2017a; Butler et al, 2019) and glucocerebrosidase (Arrant et al, 2019; Valdez et al, 2019; Zhou et al, 2019).

In the brain, PGRN is mainly expressed by neurons and microglia (Kao et al, 2017). Microglia, the resident immune cells in the brain, produce and secrete especially high levels of PGRN, especially those which have become reactive after insult or trauma (Moisse et al, 2009; Hu et al, 2010; Naphade et al, 2010; Philips et al, 2010; Zhou et al, 2017c). PGRN is a well-established modulator of immune function and has been shown to regulate microglial activation, migration, phagocytosis, complement levels, and synapse pruning (Sleegers et al, 2008; Yin et al, 2010; Pickford et al, 2011; Martens et al, 2012; Lui et al, 2016). In AD mouse models and human patients, PGRN

[1]Department of Molecular Biology and Genetics, Cornell University, Ithaca, NY, USA  [2]Department of Computational Biology, Weill Institute for Cell and Molecular Biology, Cornell University, Ithaca, NY, USA  [3]Feil Family Brain and Mind Research Institute, Weill Cornell Medicine, New York, NY, USA

Correspondence: fh87@cornell.edu

was found to be highly expressed in the microglia surrounding Aβ plaques (Pereson et al, 2009; Minami et al, 2014; Mendsaikhan et al, 2019). Microglial-specific reduction in PGRN expression has been shown to result in impaired phagocytosis, increased plaque load, and exacerbated cognitive deficits in the mouse AD models, whereas lentivirus-mediated PGRN overexpression has opposite results (Minami et al, 2014). Consistent with this result, lentivirus mediated PGRN overexpression or intrahippocampal injection of PGRN protein has been shown to reduce Aβ plaques via increasing the activity of neprilysin, a key amyloid β–degrading enzyme (Van Kampen & Kay, 2017) or through inhibiting the expression of BACE1, an enzyme involved in Aβ generation (Guan et al, 2020). However, another study has found significantly reduced diffuse Aβ plaque growth in hippocampus and cortex in aged PGRN-deficient mice using the APPswe/PS1ΔE9 (APP/PS1) mouse model (Takahashi et al, 2017). $Grn^{+/-}$ mice have also been shown to have marginally significant reduction in plaque numbers and plaque area in female mice at 16–18 mo, although limited number of mice were used in the study (Hosokawa et al, 2018). The mechanisms causing the discrepancies in these findings remain to be determined.

In addition, how PGRN regulates lysosomal function and microglia-mediated inflammation is still unclear, and whether lysosome abnormalities caused by PGRN deficiency contributes to microglial phenotypes remains to be tested. Microglia-mediated inflammation (Colonna & Butovsky, 2017; Hansen et al, 2018; Hickman et al, 2018; Aldana, 2019; Hammond et al, 2019) and endolysosomal dysfunction (Wang et al, 2018; Bajaj et al, 2019; Lie & Nixon, 2019) are common mechanisms shared by many neurodegenerative diseases. Proper endosome–lysosome function is critical for degradation of phagocytosed materials in microglia, and thus is essential for the health of microglia (Podlesny-Drabiniok et al, 2020). In this study, we show that PGRN deficiency leads to an up-regulation of lysosomal enzyme cathepsins, lysosome membrane protein CD68 and glycoprotein nonmetastatic melanoma protein B (GPNMB), cell surface receptor TREM2, as well as the β-galactoside–binding protein galectin-3 specifically in microglia near the Aβ plaque. Furthermore, we demonstrate that PGRN deficiency in microglia results in lysosome abnormalities and increased inflammatory responses after Aβ fibrils treatment. Taken together, our data support that PGRN regulates microglia-mediated inflammation through modulating lysosomal functions.

# Results

### PGRN is highly expressed in the microglia around Aβ plaques

To determine the role of PGRN in microglial responses to Aβ, we first examined PGRN localization and expression pattern in the 5XFAD mouse model, which is a rapid-onset amyloid plaque models (Oakley et al, 2006). Immunostaining showed that PGRN is mainly expressed in the IBA1-positive microglia around Aβ plaques in the 5XFAD mice (Fig S1A and B), consistent with previous reports from two other amyloid plaque mouse models, Tg2576 and APPPS1 (Pereson et al, 2009; Minami et al, 2014). In addition, PGRN exhibited strong colocalization with LAMP1 around Aβ plaques (Fig S1C), which might represent PGRN signals from the lysosome compartment in dystrophic

axons surrounding the Aβ plaques as previously reported (Gowrishankar et al, 2015).

### PGRN deficiency affects Aβ plaques deposition in the 5XFAD mice in a sex- and age-dependent manner

To determine the role of PGRN in AD pathogenesis, we crossed the 5XFAD mice with $Grn^{-/-}$ mice. The 5XFAD mice have spatial memory deficits in the Y-maze test at 4–5 mo of age (Fig S2), as reported previously (Oakley et al, 2006). However, ablation of PGRN in the 5XFAD background does not seem to have a significant effect on this behavioral change, although there was a trend of alleviating the deficits in the male mice (Fig S2).

Next, we examined whether PGRN deficiency affects Aβ plaque pathology in the 5XFAD mouse model. The 5XFAD mice start to show amyloid deposition and gliosis at 2 mo, especially in subiculum and deep cortical layers (Oakley et al, 2006). We examined Aβ pathology by using Thioflavin S (ThioS) staining to detect dense core Aβ plaques and anti-Aβ antibody (6E10) staining to quantify total Aβ plaques in the 4.5-mo-old mice. Both ThioS staining and anti-Aβ antibody staining revealed a significant reduction in the area, number and intensity of Aβ plaques in the male 5XFAD $Grn^{-/-}$ mice in the cortex compared with age-matched male 5XFAD mice (Fig 1A and B). A similar trend was observed in the subiculum region but did not reach statistical significance (Fig 1B). On the other hand, ablation of PGRN does not seem to affect Aβ plaque deposition in the female 5XFAD mice (Fig 1). To further dissect the role of PGRN in regulating Aβ dynamics, we performed ELISA to measure the levels of soluble and insoluble Aβ in the cortical lysates. PGRN deficiency in male mice results in a significant decrease of Aβ40 and Aβ42 in both soluble and insoluble fractions (Fig 2A). No effect of PGRN ablation was observed in the female mice (Fig 2A), consistent with our immune-staining results (Fig 1).

The decrease in the Aβ levels in both the soluble and insoluble fractions suggests that Aβ production might be affected by PGRN loss in males. Thus, we decided to examine the levels of amyloid precursor protein (APP) in these mice. Immunoblot analysis revealed a significant reduction in the levels of mutant human APP expressed from the transgene in the 4.5-mo-old male mice (Fig 2B and C). No significant changes in the levels of presenilin1 (PS1), BACE1, or endogenous mouse APP have been observed in these mice (Fig 2B and C). Since both APP and PS1 transgenes are driven by the mouse Thy1 promoter in the 5XFAD mice and PGRN deficiency only affects APP levels, it suggests that PGRN affects APP levels post-transcriptionally. However, PGRN deficiency alone do not have any effect on the levels of endogenous APP, PS1, or BACE1 in mice (Fig S3). These data suggests that loss of PGRN specifically affects in the metabolism of mutant human APP protein expressed from the transgene in the young male mice.

Because Aβ pathology is a dynamic process and worsens with aging, we also analyzed Aβ accumulation in the 10-mo-old 5XFAD and 5XFAD $Grn^{-/-}$ mice. ThioS staining and anti-Aβ antibody staining showed either no significant change or a slight decrease in the area and number of Aβ plaques in the both male and female 5XFAD $Grn^{-/-}$ mice compared with age- and sex-matched 5XFAD mice (Figs 3A and S4). The total intensity of Aβ plaques and the levels of both soluble and insoluble Aβ40 and Aβ42 are not altered in both male

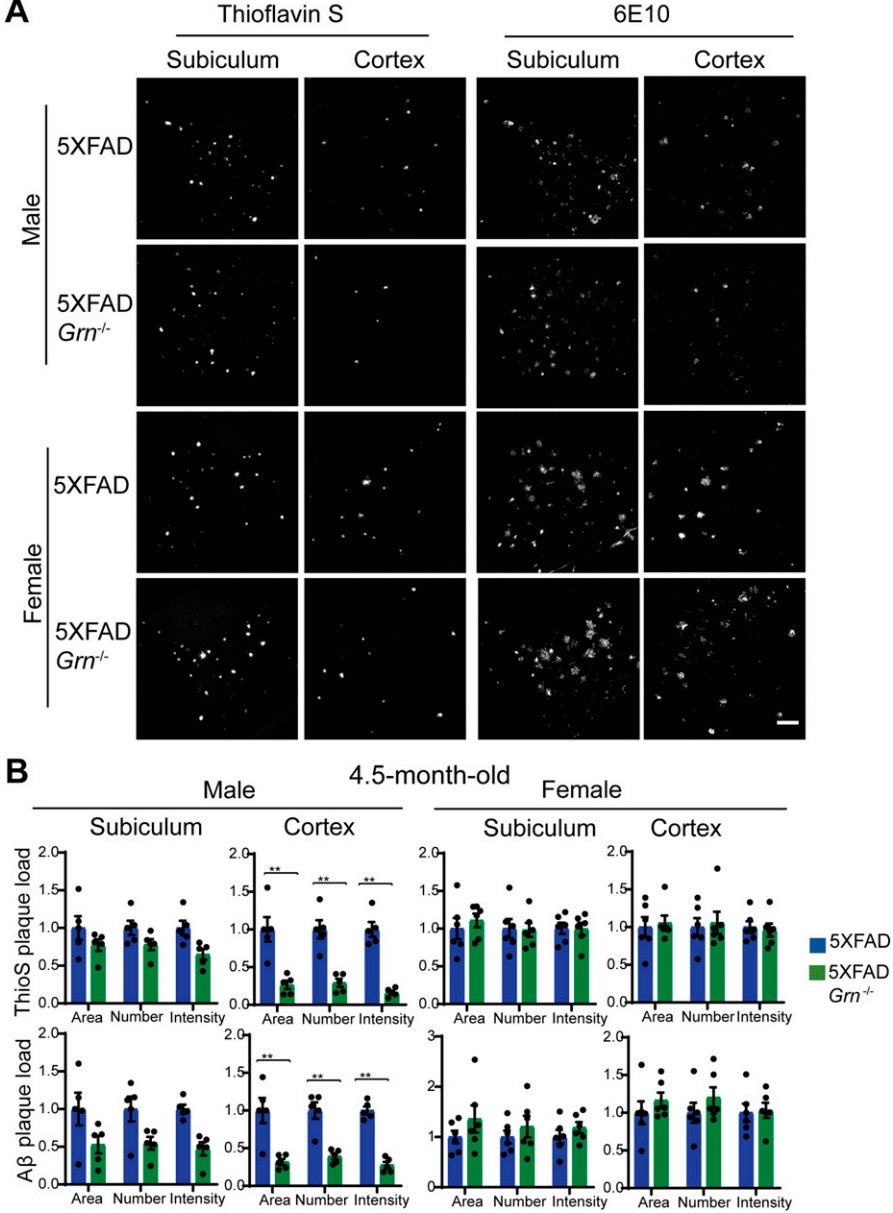

**Figure 1. PGRN deficiency affects Aβ levels in the 4.5-mo-old 5XFAD mice in a sex-specific manner.**
**(A)** Representative images of Thioflavin S (ThioS) staining and Aβ (6E10) immunostaining of subiculum and frontal cortex of 4.5-mo-old male and female 5XFAD, 5XFAD *Grn*$^{-/-}$ mice. Scale bar = 100 μm. **(B)** Quantification of ThioS-positive and Aβ immunoreactive area, number, and intensity in subiculum and cortex of 4.5-mo-old male and female 5XFAD, 5XFAD *Grn*$^{-/-}$ mice. Mean ± SEM, n = 5–6, *t* test, **$P < 0.01$.

and female *Grn*$^{-/-}$ mice (Fig 3A and B). No difference in APP levels was detected between 5XFAD and 5XFAD *Grn*$^{-/-}$ mice at this age (data not shown). Taken together, our results indicate a specific role of PGRN in regulating mutant human APP levels in the young male mice and a modest role of PGRN in regulating the dynamics of Aβ plaques but not total Aβ levels at older ages.

### PGRN deficiency results in enhanced microglial activation near the Aβ plaques

To dissect molecular pathways affected by the loss of PGRN, we performed RNA-Seq analysis of hippocampus samples from 4.5-mo-old WT, *Grn*$^{-/-}$, 5XFAD, and 5XFAD *Grn*$^{-/-}$ mice. Loss of PGRN alone results in minimal changes in gene expression at this young

age, but loss of PGRN in the 5XFAD background leads to differential expression of many genes in the hippocampus (Supplemental Data 1 and Fig S5). Among the 14 genes up-regulated in the 5XFAD *Grn*$^{-/-}$ hippocampus samples, inflammation and lysosome pathways are identified (Fig S5A and B).

Many genes up-regulated in 5XFAD *Grn*$^{-/-}$ samples belong to the gene signature associated with microglia activation during neurodegeneration, so called diseased associated microglia (DAM) or neurodegenerative microglia (mGnD) (Krasemann et al, 2017; Deczkowska et al, 2018), including *Lyz2*, *Tyrobp*, *Lgals3*, *Cd68*, and glycoprotein nonmetastatic melanoma protein B (*Gpnmb*) (Fig S5). *Lgals3* (encoding galectin-3 protein) and *Gpnmb* are two most up-regulated genes in the 5XFAD *Grn*$^{-/-}$ samples (Fig S5B), consistent with a recent proteomic study identifying galectin-3 and

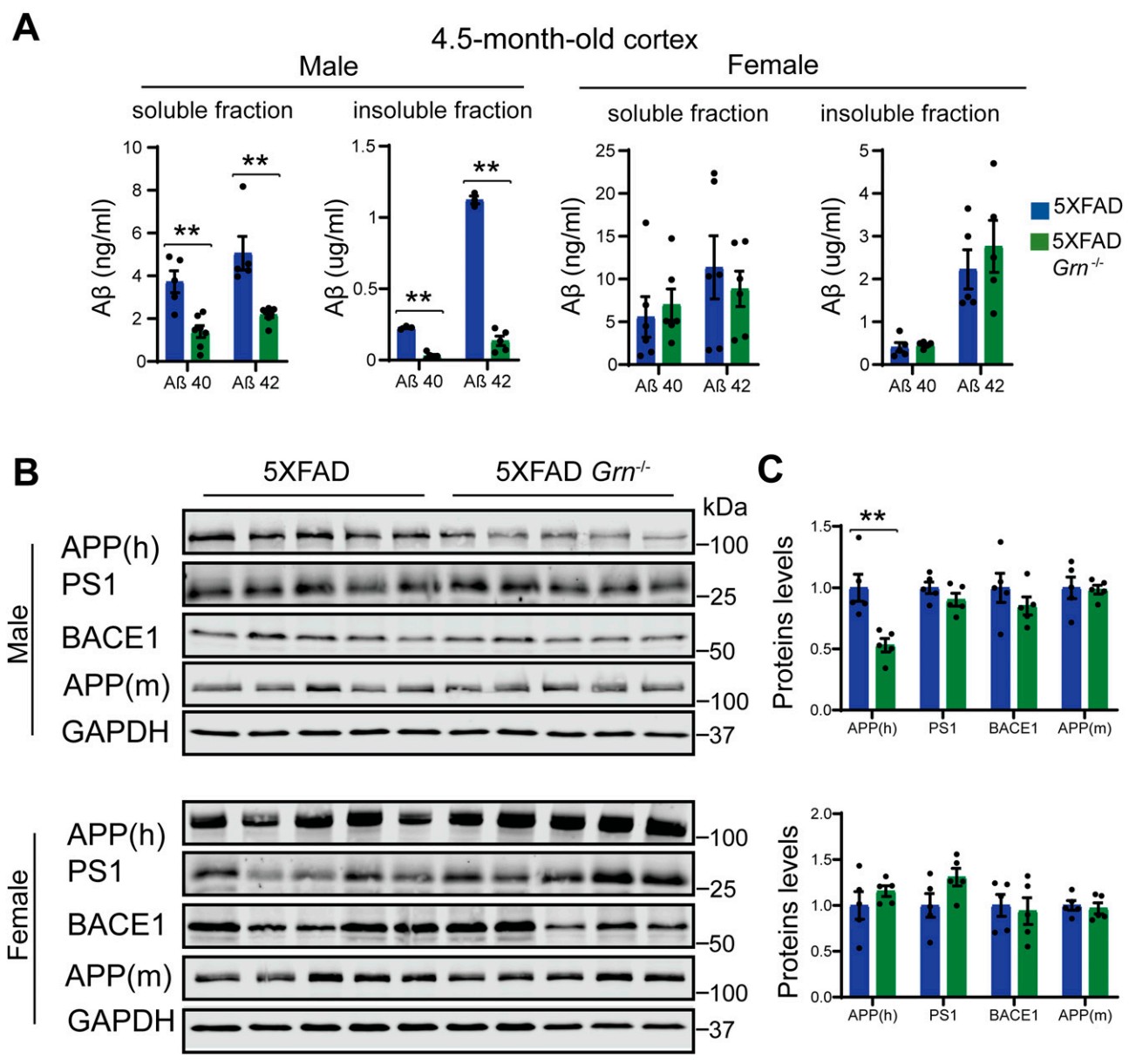

**Figure 2. PGRN deficiency decreases the levels of Aβ and human amyloid precursor protein (APP) in the 4.5-mo-old male 5XFAD mice.**
**(A)** Aβ40 and Aβ42 levels in the RIPA-soluble fraction (soluble) and guanidine-soluble fraction (insoluble) from 4.5-mo-old male and female 5XFAD, 5XFAD $Grn^{-/-}$ mice were measured by ELISA. Mean ± SEM, n = 3–6, $t$ test, **$P$ < 0.01. **(B, C)** Western blot analysis of APP, presenilin1 (PS1), and BACE1 in the soluble fractions of cortical lysates from 4.5-mo-old male and female 5XFAD, 5XFAD $Grn^{-/-}$ mice. 6E10 antibody was used to specifically detect human APP and a rabbit polyclonal antibody was used to detect mouse APP. The proteins levels were quantified and normalized to GAPDH. Mean ± SEM, n = 5, $t$ test, **$P$ < 0.01.

GPNMB as the two proteins most up-regulated in response to PGRN loss (Huang et al, 2020). The up-regulation of galectin-3 and GPNMB was confirmed by Western blot analysis of brain lysates. Both galectin-3 and GPNMB show a specific up-regulation in the 4.5- and 10-mo-old 5XFAD $Grn^{-/-}$ brain lysates, with minimal changes in the 5XFAD or $Grn^{-/-}$ samples (Figs 4A–D and 5A–D), supporting that the simultaneous loss of PGRN and Aβ stimulation result in excessive microglia activation. Although the male mice show significantly reduced numbers of Aβ plaques, immunostaining shows a significant up-regulation of galectin-3 in the IBA1-positive microglia near Aβ plaques in both male and female mice (Fig 4E and F),

indicating that PGRN does not have a sex-specific effect regarding microglial responses to Aβ plaques. GPNMB expression is also specifically up-regulated in the microglia near the Aβ plaques in the absence of PGRN (Fig 5E and F). In addition, an overlap in the expression pattern of galectin-3 and GPNMB is observed (Fig 5G), suggesting that these two genes might be subject to regulation by the same pathway. Next, we examined whether levels of TREM2, a master regulation of microglia homeostasis (Deczkowska et al, 2020), are affected in 5XFAD $Grn^{-/-}$ mice. Co-staining of IBA1, TREM2, and ThioS indicates PGRN deficiency results in increased TREM2 levels in the microglia near the Aβ plaques (Fig 6A and B). Taken together,

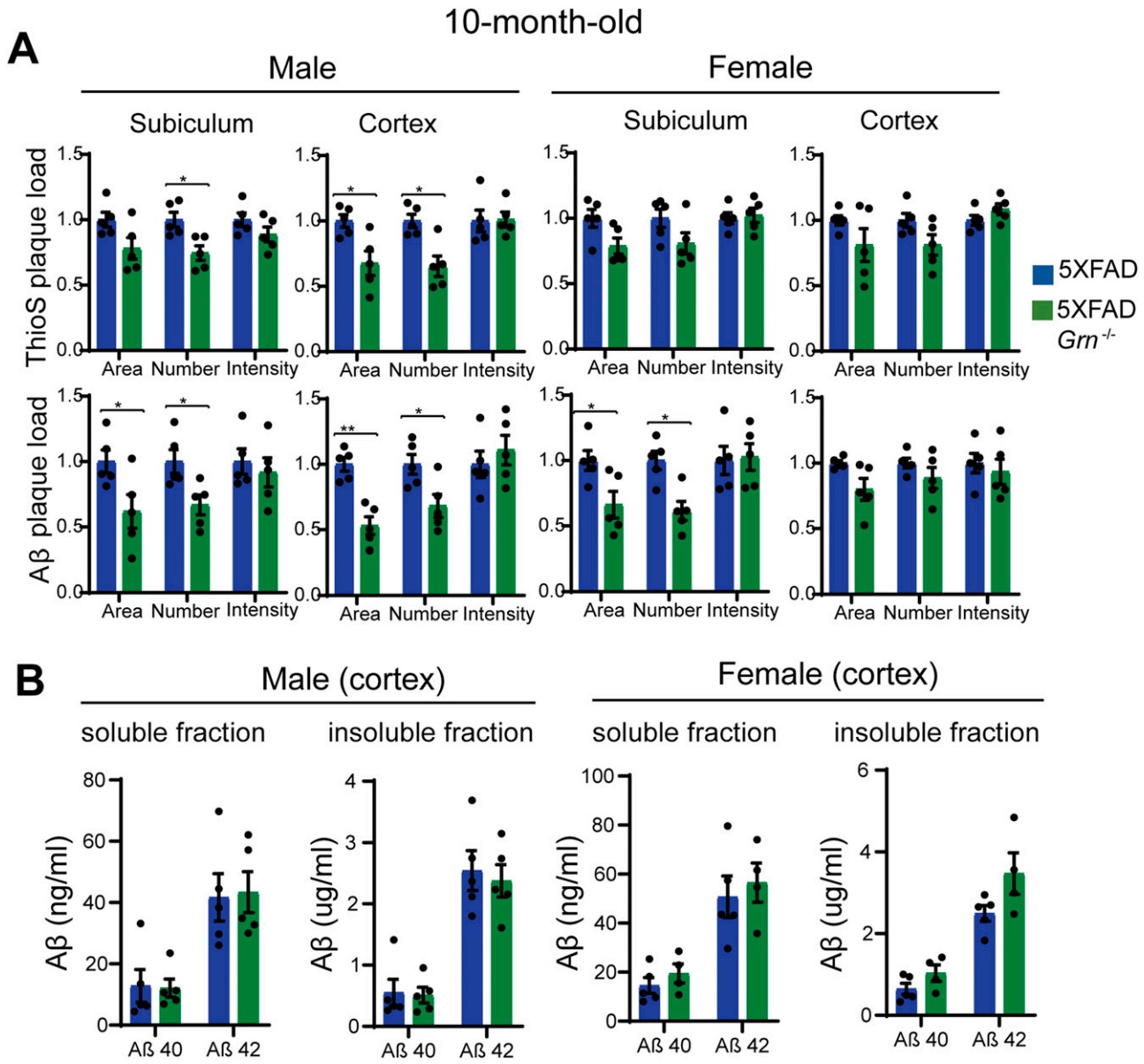

**Figure 3. The effect of PGRN on Aβ plaque in the 10-mo-old 5XFAD mice.**
**(A)** Quantification of ThioS-positive and Aβ immunoreactive area, number and intensity in subiculum and cortex of 10-mo-old male and female 5XFAD, 5XFAD $Grn^{-/-}$ mice. Mean ± SEM, n = 5, $t$ test, *$P < 0.05$, **$P < 0.01$. **(B)** Aβ40 and Aβ42 levels in the RIPA-soluble fraction (soluble) and guanidine-soluble fraction (insoluble) from 10-mo-old male and female 5XFAD, 5XFAD $Grn^{-/-}$ mice were measured by ELISA. Mean ± SEM, n = 4–5, $t$ test.

these results indicate that several DAM proteins, including galectin-3, GPNMB, and TREM2, are up-regulated in microglia around Aβ plaques in response to PGRN loss.

Loss of PGRN alone is known to cause microglial activation in an age-dependent manner (Kao et al, 2017). Consistently, immunostaining of IBA1, a microglia marker, and CD68, a marker for activated microglia also showed enhanced microglia activation in the cortex region in both male and female PGRN-deficient mice at 10-mo of age (Fig 6C and D). However, this difference is much more pronounced in the 5XFAD $Grn^{-/-}$ mice compared with age-matched 5XFAD mice (Fig 6C and D). In addition, an enrichment of CD68 signals is observed in the microglia near Aβ plaques in the PGRN-deficient mice (Fig 6C and

E). These data further support that Aβ stimulation exacerbates microglial activation in the absence of PGRN.

**Increased levels of lysosome proteins near the Aβ plaques in the PGRN-deficient mice**

Because PGRN deficiency is known to cause lysosome dysfunction (Paushter et al, 2018), we next examined alterations in lysosome proteins in 4.5- and 10-mo-old 5XFAD mice with and without PGRN expression. The levels of several lysosome proteins including cathepsin B (CathB), cathepsin D (CathD), cathepsin L (CathL), LAMP1, and LAMP2 were significantly increased in the brain lysates from 5XFAD

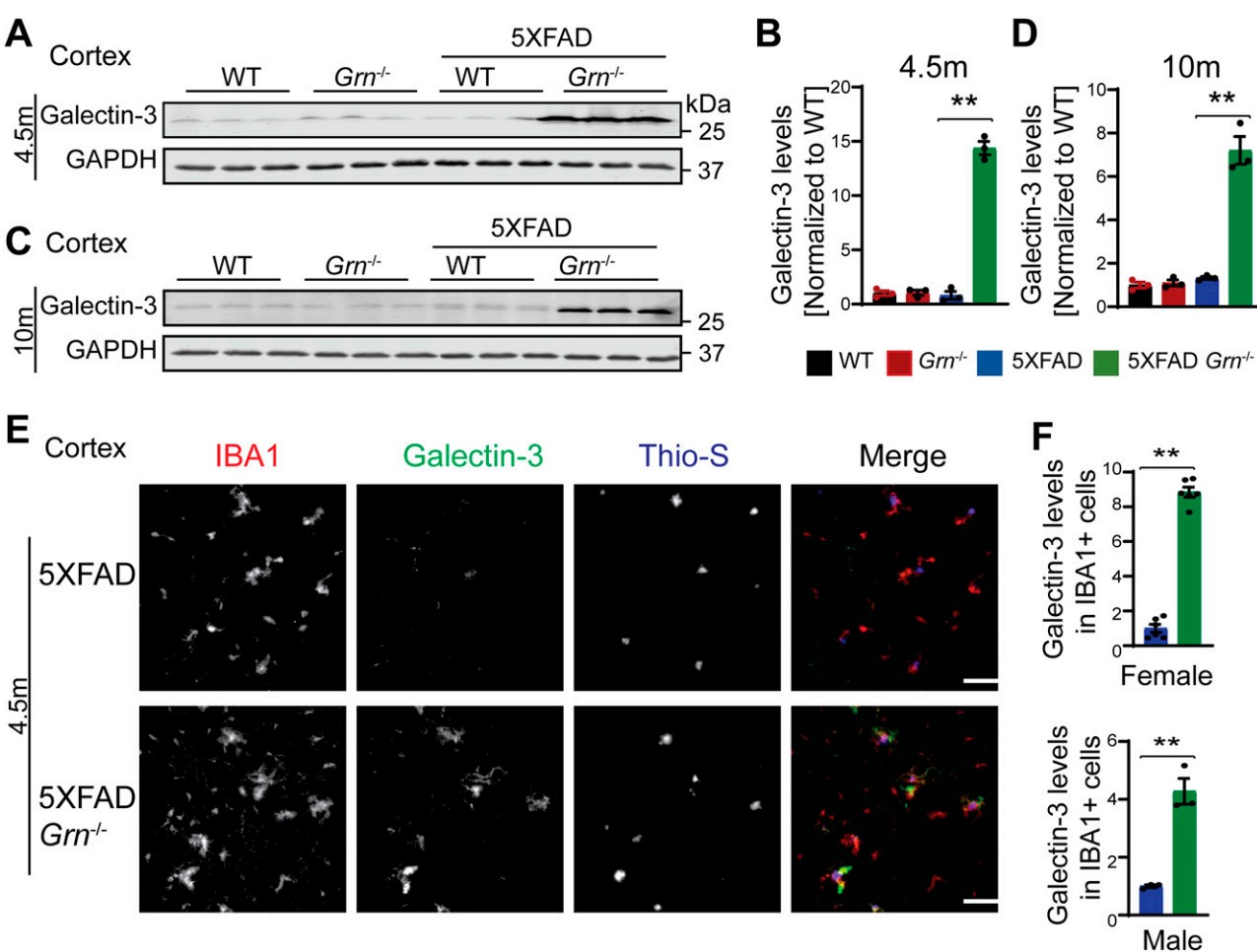

**Figure 4. PGRN deficiency leads to an increase in the levels of galectin-3 in the microglia surrounding the Aβ plaque in the 5XFAD mice.**
**(A, B, C, D)** Western blot analysis of galectin-3 in the cortical lysates from 4.5-mo-old (A, B) and 10-mo-old (C, D) WT, $Grn^{-/-}$, 5XFAD and 5XFAD $Grn^{-/-}$ mice. Mean ± SEM; n = 3, one-way ANOVA, **$P < 0.01$. Female 4.5-mo-old 5XFAD and 5XFAD $Grn^{-/-}$ mice were used. For other groups, mice of mixed sex were used. **(E)** Immunostaining of IBA1, galectin-3, and ThioS in the brain sections from 4.5-mo-old female 5XFAD and 5XFAD $Grn^{-/-}$ mice. Representative images from the cortex region were shown. Similar phenotypes were observed in the male 5XFAD and 5XFAD $Grn^{-/-}$ mice, although with reduced numbers of plaques. Scale bar = 100 μm. **(E, F)** Quantification of galectin-3 levels in IBA1-positive microglia in the cortex region for experiment in (E). Mean ± SEM; n = 3–6, $t$ test, **$P < 0.01$.
Source data are available for this figure.

$Grn^{-/-}$ mice compared with 5XFAD mice, at both 4.5 and 10 mo of age, with more pronounced differences at 10 mo old (Figs 7A and B, S6A and B, and S7), suggesting a progressive lysosomal response to Aβ-accumulation and PGRN loss. These changes were not observed in the 5XFAD or $Grn^{-/-}$ mice, supporting that the combination of PGRN loss and Aβ stimulation drives the expression changes of lysosome genes.

Next, we examined which cell types up-regulate the expression of lysosome genes in 5XFAD $Grn^{-/-}$ mice. Co-staining of 6E10 and CathB, with microglia markers IBA1 indicates significant up-regulation of the lysosome protease CathB in microglia in the brain sections of 5XFAD $Grn^{-/-}$ mice, especially around Aβ plaques (Fig 8A–C). Immunostaining with antibodies against IBA1, galectin-3, and CathB shows galectin-3 and CathB are co-regulated in the microglia in 5XFAD $Grn^{-/-}$ mice (Fig 8D). In addition, the lysosome protease CathD is abnormally up-regulated in the CD68 positive microglia around Aβ plaques in the both male and female 5XFAD $Grn^{-/-}$ mice (Fig 9A–C), supporting that PGRN is critical for maintaining lysosome homeostasis in microglia in response to Aβ stimulation.

Many of the lysosome genes are under the transcriptional control of the members of melanocyte inducing transcription factor (MiTF) family, including MiTF, TFEB, and TFE3. Under starvation or lysosome stress, these transcriptional factors are translocated into the nucleus to induce the expression of many genes in the autophagy lysosome pathway (Napolitano & Ballabio, 2016; Bajaj et al, 2019; Saftig & Puertollano, 2021). To determine whether this pathway is activated in the 5XFAD $Grn^{-/-}$ mice, we stained brain sections with antibodies against TFE3 and IBA1. A significant increase in nuclear TFE3 signals was observed in the microglia in the brain sections of 5XFAD $Grn^{-/-}$ mice, compared with 5XFAD controls (Fig 9D and E). These data suggest that lysosome abnormalities in microglia results in a feedback loop through the TFE3/TFEB pathway to up-regulate the expression of lysosome genes and possibly also inflammation genes. Interestingly, activation of TFE3/TFEB has been shown to drive the expression of inflammation genes in several studies (Nabar & Kehrl, 2017; Brady et al, 2018). However, we did not detect a direct correlation between the increase in TFE3 nuclear

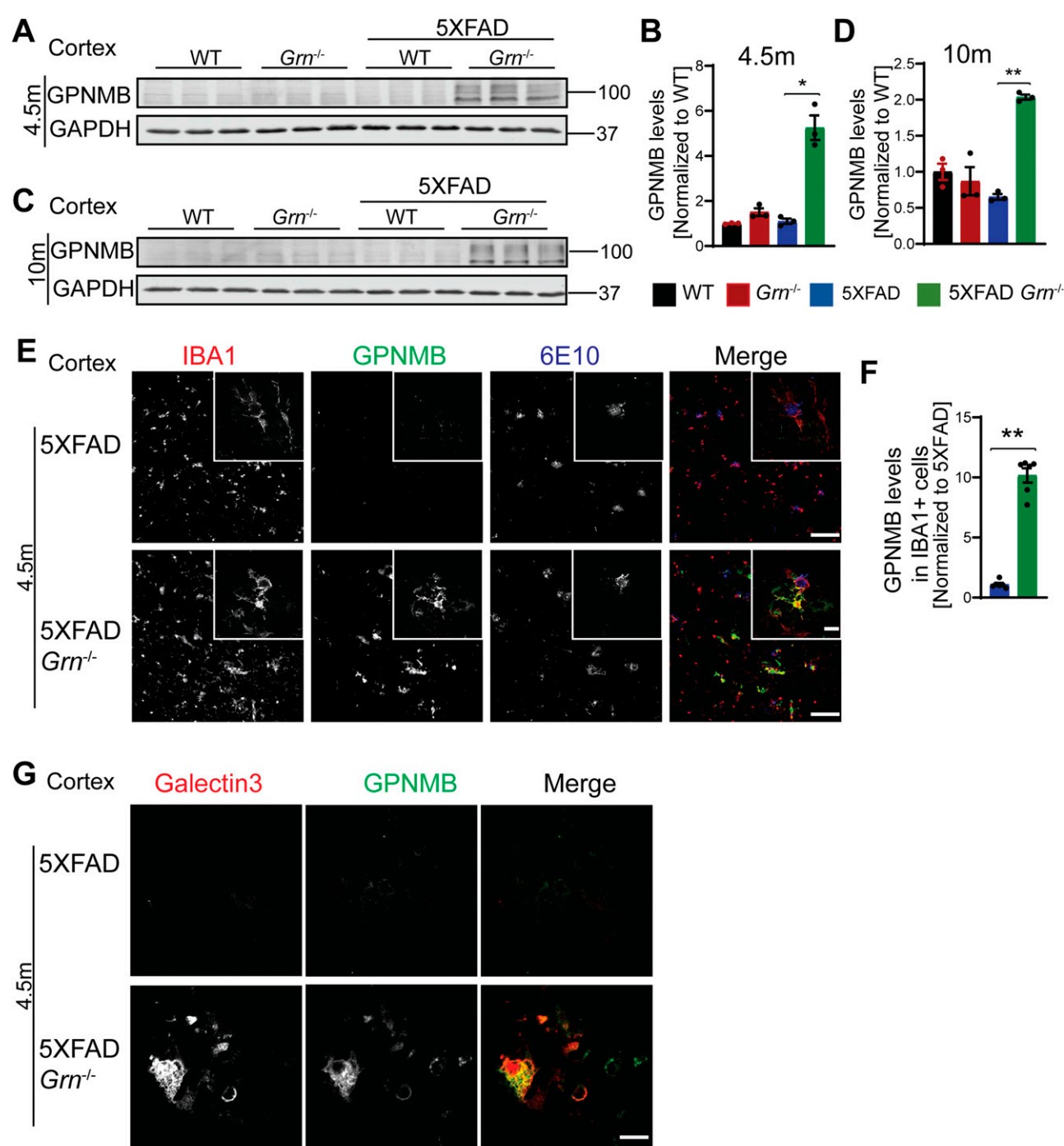

**Figure 5. PGRN deficiency leads to an increase in the levels of microglial glycoprotein nonmetastatic melanoma protein B (GPNMB) in the 5XFAD mice.**
**(A, B, C, D)** Western blot analysis of GPNMB in the cortical lysates from 4.5-mo-old (A, B) and 10-mo-old (C, D) WT, $Grn^{-/-}$, 5XFAD and 5XFAD $Grn^{-/-}$ mice. Female 4.5-mo-old 5XFAD and 5XFAD $Grn^{-/-}$ mice were used. For other groups, mice of mixed sex were used. Mean ± SEM; n = 3, one-way ANOVA, $*P < 0.05$, $**P < 0.01$. The same samples were used for experiments in Figs 4A–D and 5A–D and the experiments were performed simultaneously. **(E)** Immunostaining of IBA1, GPNMB, and A$\beta$ (6E10) in the brain sections from 4.5-mo-old female 5XFAD and 5XFAD $Grn^{-/-}$ mice. Representative images from the cortex region were shown. Scale bar = 100 μm. Representative high-resolution confocal images were shown in inset. Scale bar = 10 μm. **(E, F)** Quantification of GPNMB levels in IBA1-positive microglia in the cortex region for experiment in (E). Mean ± SEM; n = 6, $t$ test, $**P < 0.01$. **(G)** Representative confocal high-resolution images of GPNMB and galectin-3 co-staining in the brain sections from 4.5-mo-old 5XFAD and 5XFAD $Grn^{-/-}$ mice. Scale bar = 10 μm.

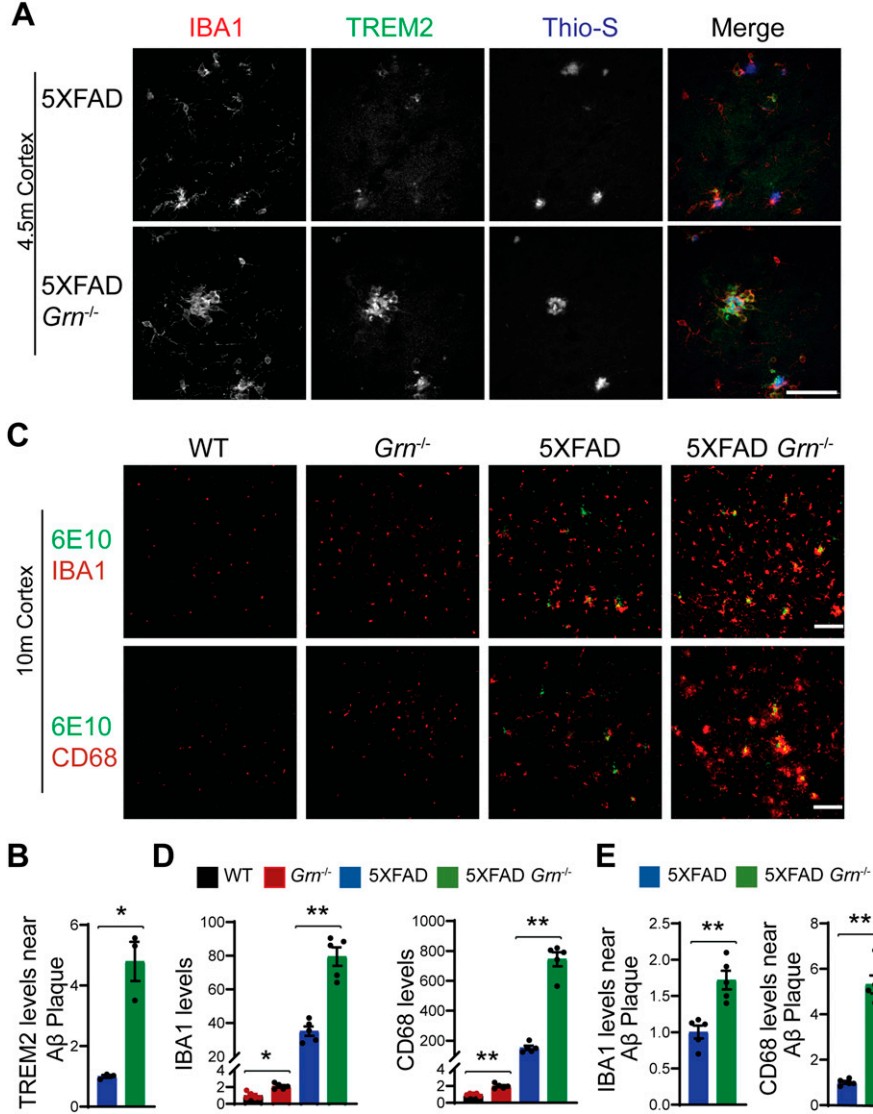

**Figure 6. PGRN deficiency leads to increase microglial activation around the Aβ plaques in the 5XFAD mice.**

**(A)** Representative confocal high-resolution images of IBA1, TREM2, and ThioS co-staining in the brain sections from 4.5-mo-old female 5XFAD and 5XFAD *Grn*⁻/⁻ mice. Scale bar = 50 μm. **(A, B)** Quantification of TREM2 signals near Aβ plaques for experiment in (A). Mean ± SEM; n = 3, *t* test, *P < 0.05. **(C)** Brain sections from 10-mo-old WT, *Grn*⁻/⁻, 5XFAD and 5XFAD *Grn*⁻/⁻ mice were stained with anti-Aβ (6E10) and anti-IBA1 or anti-CD68 antibodies as indicated. Representative images from the cortex region were shown. Scale bar = 100 μm. **(C, D)** Quantification of IBA1 and CD68 signals in the cortex region for experiment in (C). Mixed female and male mice were used for the study. Mean ± SEM; n = 5, one-way ANOVA, *P < 0.05, **P < 0.01. **(C, E)** Quantification of IBA1 and CD68 signals near the Aβ plaque in the cortex region for experiment in (C). Mean ± SEM; n = 5, *t* test, **P < 0.01.

staining intensity and the up-regulation of galectin-3 expression in the microglia near the plaque, indicating that there are other transcription factors involved in regulating galectin-3 expression.

### Aβ treatment leads to enhanced inflammation and lysosomal abnormalities in PGRN-deficient microglia

To further dissect the connection between lysosome abnormalities and inflammatory responses in PGRN-deficient microglia, we examined the role of PGRN in lysosomal and inflammatory responses in cell culture. We deleted PGRN in the macrophage cell line Raw 264.7 using the CRISPR/Cas9 technique and examined the lysosomal membrane permeabilization (LMP) using galectin-3 as a marker. Upon LMP, intracellular galectin-3 has been shown to be recruited to damaged lysosomes, forming discernible puncta (Aits et al, 2015). L-leucyl-L-leucine methyl ester (LLOME), a lysosome-damaging reagent was used to induce LMP. A significantly increased

overlap of galectin-3 and LAMP1 was observed in PGRN-deficient Raw 264.7 cells compared with control cells upon LLOME treatment (Fig 10A and B). Similar results were obtained from primary microglia cultured from WT and PGRN-deficient mice, indicating that PGRN deficiency affects lysosome membrane integrity (Fig 10C and D).

Next, we examined the effect of Aβ fibril treatment on lysosome changes in WT and *Grn*⁻/⁻ primary microglia. A significant increase in the levels of galectin-3, cathepsin D and GPNMB was observed in PGRN-deficient primary microglia (Figs 11A and B and 12A and B). To determine whether the transcription factor TFE3 is involved in response to Aβ fibrils, we stained primary microglia with antibodies against TFE3. A significant increase in the ratio of nuclear to cytoplasmic TFE3 signals was observed in PGRN-deficient primary microglia, compared with WT controls (Fig 12C and D). This result indicates that Aβ fibril treatment results in lysosome abnormalities in *Grn*⁻/⁻ microglia, which results in activation of the TFE3/TFEB pathway

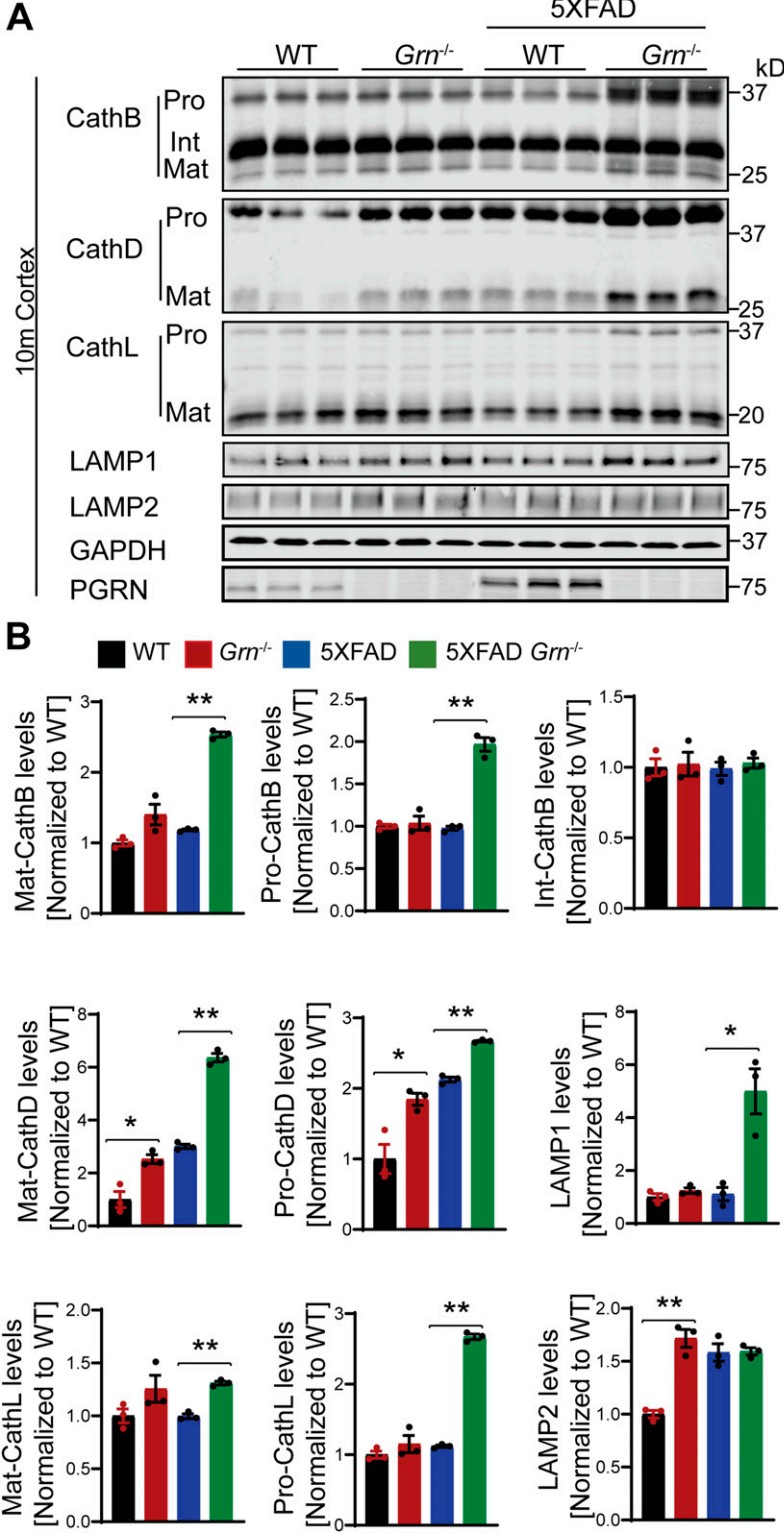

**Figure 7. PGRN deficiency leads to an increase in the levels of lysosome proteins in the 5XFAD mice.**
**(A)** Cortical lysates from 10-mo-old WT, $Grn^{-/-}$, 5XFAD and 5XFAD $Grn^{-/-}$ mice were subjected to Western blot with anti-cathepsin B (CathB), CathD, CathL, LAMP1, LAMP2, PGRN, and GAPDH antibodies. Mixed female and male mice were used for the study. **(A, B)** Quantifications of lysosome proteins levels for experiment in (A). Mean ± SEM; n = 3, one-way ANOVA, *P < 0.05, **P < 0.01.

to up-regulate the expression of lysosome genes. Next, we assessed whether the lysosome abnormalities caused by PGRN deficiency contributed to enhanced inflammatory responses. Aβ fibril treatment leads to a significant increase in the levels of the proinflammatory marker TNF-α in the $Grn^{-/-}$ Raw macrophages and microglia compared with WT microglia (Fig 12E and F). Taken together, our studies support that Aβ fibril treatment leads to enhanced inflammation and lysosomal responses in PGRN-deficient microglia.

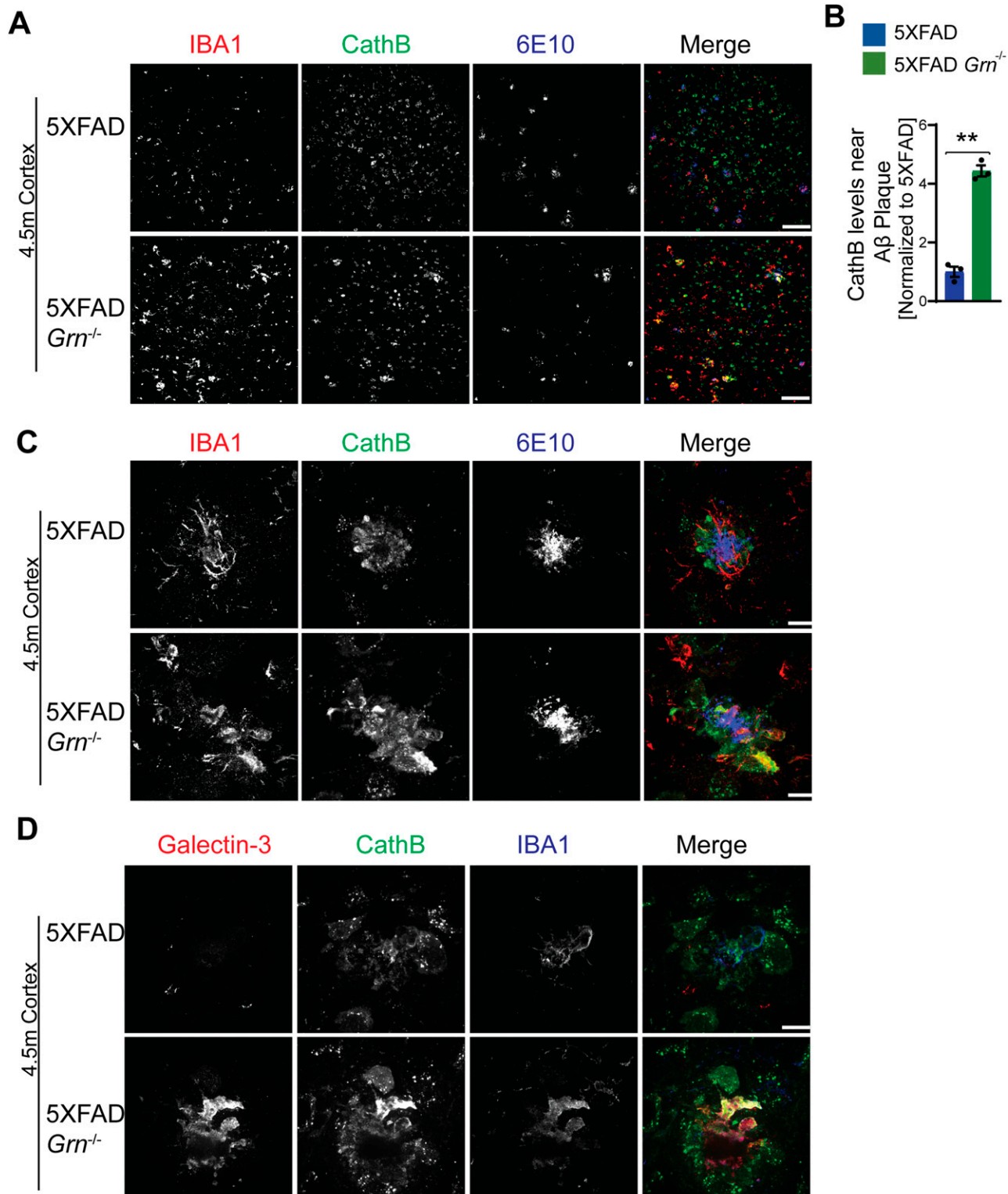

**Figure 8. PGRN deficiency leads to an up-regulation of cathepsin B in the microglia near Aβ plaque in the 5XFAD mice.**
**(A)** Immunostaining of IBA1, cathepsin B (CathB), and Aβ (6E10) in the brain sections from 4.5-mo-old female 5XFAD and 5XFAD $Grn^{-/-}$ mice. Representative images from the cortex region were shown. Scale bar = 100 μm. **(A, B)** Quantification of CathB levels near Aβ plaque for experiment in (A). Mean ± SEM; n = 3, *t* test, **P < 0.01. **(C)** Representative confocal high-resolution images of IBA1, CathB, and Aβ (6E10) co-staining in the brain sections from 4.5-mo-old 5XFAD and 5XFAD $Grn^{-/-}$ mice. Scale bar = 10 μm. **(D)** Representative confocal high-resolution images of galectin-3, CathB, and IBA1 co-staining in the brain sections from 4.5-mo-old 5XFAD and 5XFAD $Grn^{-/-}$ mice. Scale bar = 10 μm.

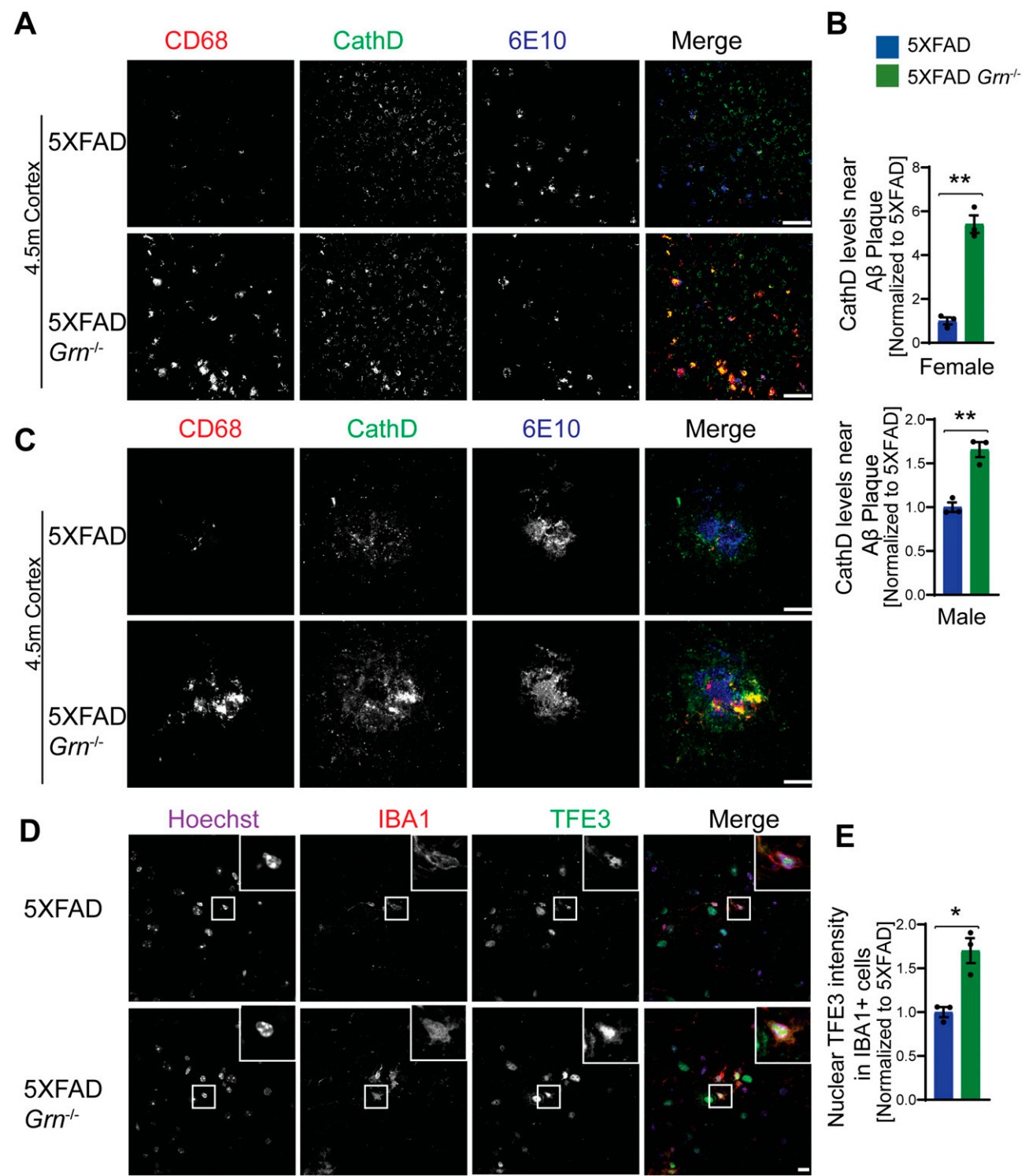

**Figure 9. PGRN deficiency results in increased cathepsin D expression and TFE3 nuclear translocation in the microglia near the Aβ plaque in the 5XFAD mice.**
**(A)** Immunostaining of CD68, CathD, and Aβ (6E10) in the brain sections from 4.5-mo-old female 5XFAD and 5XFAD *Grn*⁻/⁻ mice. Representative images from the cortex region were shown. Similar phenotypes were seen in the male 5XFAD and 5XFAD *Grn*⁻/⁻ mice, although with reduced numbers of plaques. Scale bar = 100 μm. **(A, B)** Quantification of CathD levels near Aβ plaque for experiment in (A). Mean ± SEM; n = 3, *t* test, **P < 0.01. **(C)** Representative confocal high-resolution images of CD68, CathD, and Aβ (6E10) co-staining in the brain sections from 4.5-mo-old 5XFAD and 5XFAD *Grn*⁻/⁻ mice. Scale bar = 10 μm. **(D)** Immunostaining of IBA1, TFE3 in the brain sections from 4.5-mo-old 5XFAD and 5XFAD *Grn*⁻/⁻ mice. Representative images from the cortex region were shown. Scale bar = 10 μm. **(D, E)** Nuclear signals of TFE3 in IBA1-positive microglia were quantified for experiment in (D). Mean ± SEM; n = 3, *t* test, *P < 0.05.

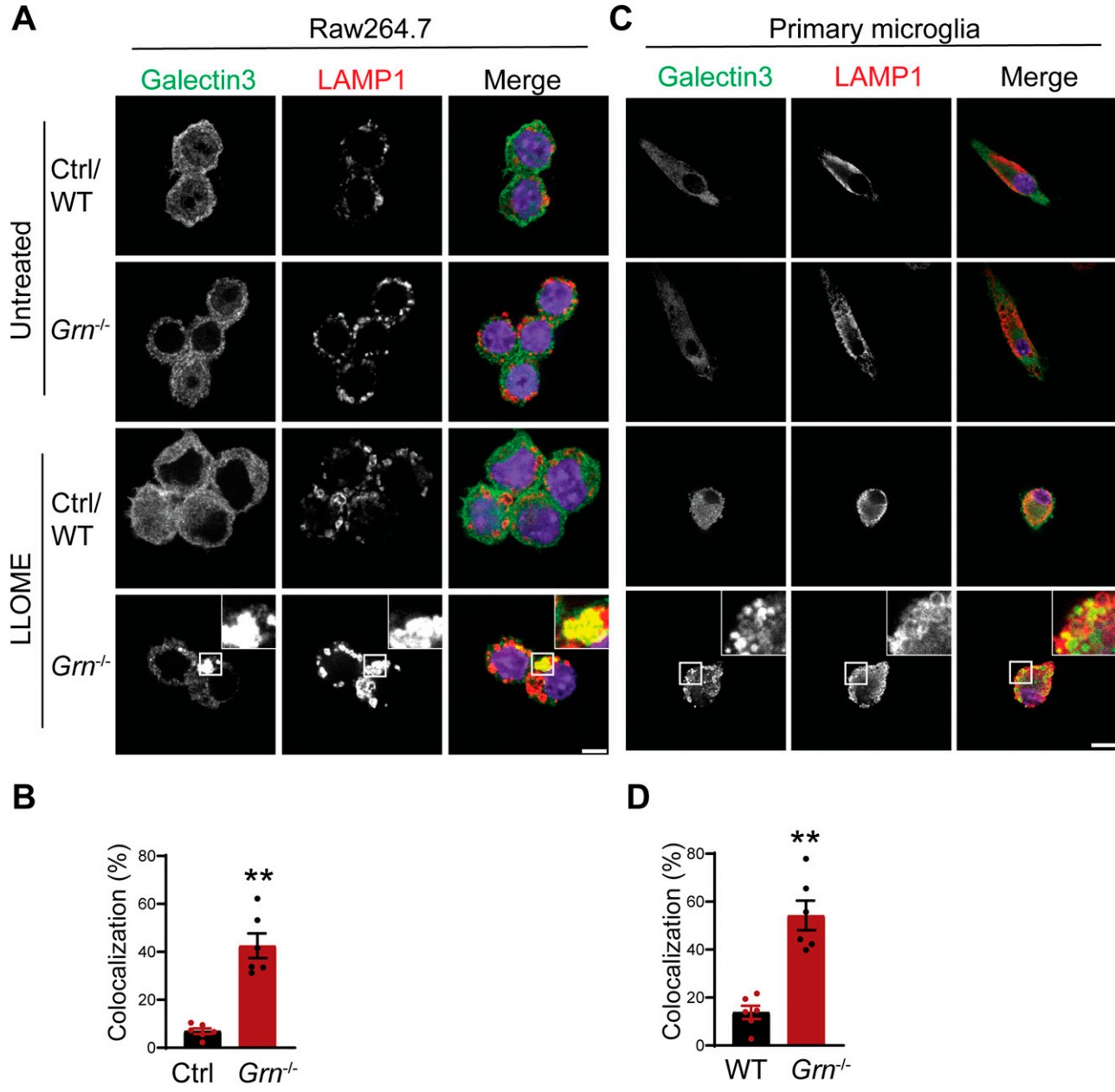

**Figure 10. PGRN-deficient macrophage and microglia are more prone to lysosomal membrane leakage.**
**(A)** Representative confocal images of Raw 264.7 cells treated with 500 μM LLOME for 2 h and stained with rat anti-LAMP1 (red) and mouse anti–galectin-3 antibodies (green). Scale bar = 10 μm. **(A, B)** Manders' coefficients for colocalization of galectin-3 with LAMP1 for experiment in (A). Mean ± SEM; n = 6, *t* test, **P < 0.01.
**(C)** Representative confocal images of primary microglia treated with 500 μM LLOME for 2 h and stained with rat anti-LAMP1 (red) and mouse anti–galectin-3 antibodies (green). Scale bar = 10 μm. **(C, D)** Manders' coefficients for colocalization of galectin-3 with LAMP1 for experiment in (C). Mean ± SEM; n = 6, *t* test, **P < 0.01.

# Discussion

### A sex-and-age specific role of PGRN in regulating human APP metabolism in mice

Although PGRN has been implicated in AD, the exact role of PGRN in AD disease progression remains unclear. Conflicting results have been obtained regarding the effect of PGRN in AD in mouse models. PGRN deficiency has been demonstrated to exacerbate pathological

phenotypes, including Aβ plaque burden, neuroinflammatory markers, and memory impairment in AD mouse models in some studies (Minami et al, 2014; Van Kampen & Kay, 2017). Other studies, however, have demonstrated that some AD phenotypes are improved by PGRN deficiency (Takahashi et al, 2017; Hosokawa et al, 2018).

In this study, we examined the role of PGRN in Aβ dynamics using the 5XFAD mice which express human APP and presenilin1 transgenes containing five AD-linked mutations (APP K670N/M671L [Swedish] + I716V [Florida] + V717I [London] and PS1 M146L +

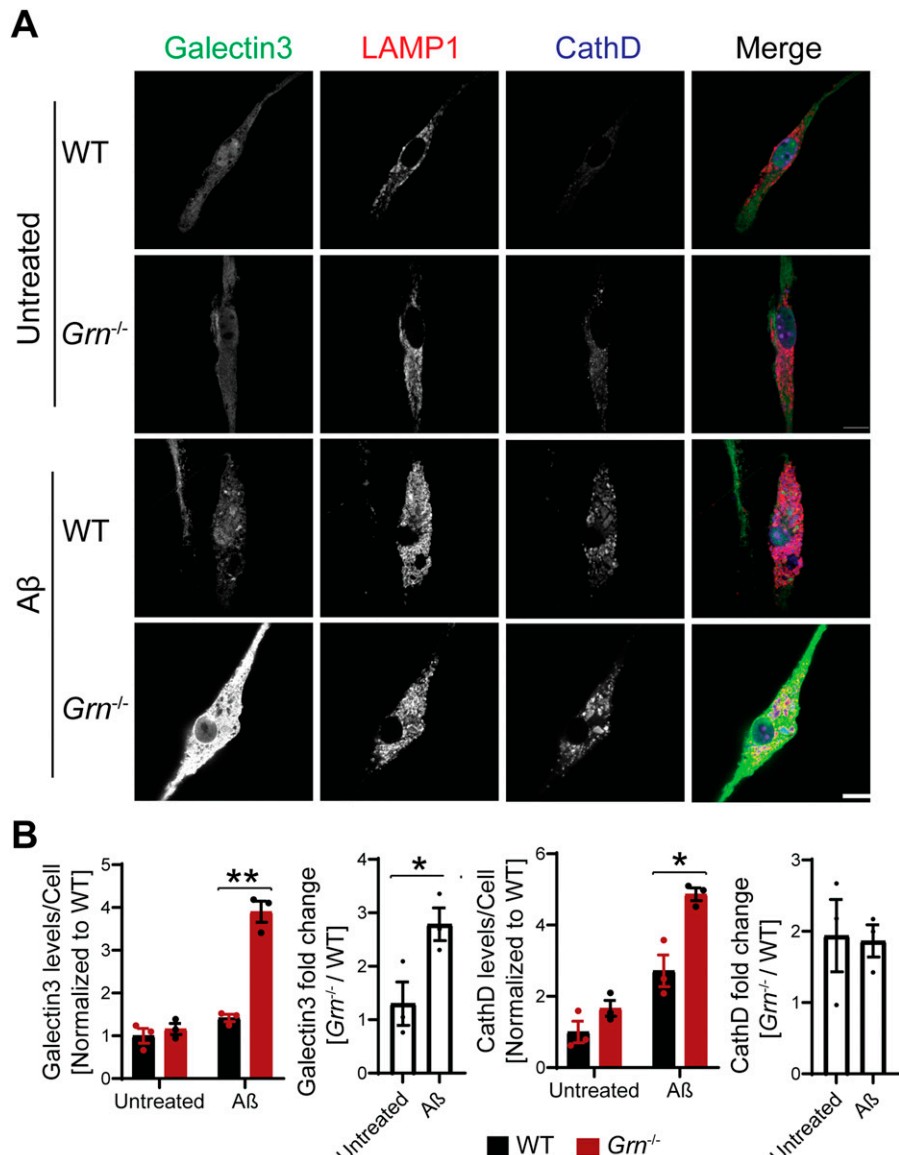

**Figure 11. PGRN-deficient microglia are more susceptible to lysosome abnormalities in response to Aβ fibrils.**
**(A)** Representative confocal images of primary microglia treated with 10 μM Aβ fibrils for 24 h and stained with anti–galectin-3, anti-LAMP1, and anti-CathD antibodies as indicated. Scale bar = 10 μm. **(A, B)** Quantification of galectin-3 and CathD levels per cell for experiment in (A). Mean ± SEM; n = 3, two-way ANOVA, *P < 0.05, **P < 0.01. **(A)** Quantification of galectin-3 and CathD fold changes ($Grn^{-/-}$/WT) for experiment in (A): t test, *P < 0.05.

L286V). We found that PGRN deficiency results in decreased Aβ plaque accumulation in the young males but not in females (Figs 1 and 2). Furthermore, we demonstrated that the reduction in Aβ plaques in the 4.5-mo-old male mice is due to a reduction in the levels of human mutant APP (Fig 2B and C). The nature of this sex dimorphic regulation of mutant APP by PGRN is unclear at this moment, but the sex-specific role of PGRN has been reported. Genetic variability in the *GRN* gene has been shown to influence the risk for developing AD in a male-specific manner in Finnish population (Viswanathan et al, 2009), which might be explained by the results from our study. PGRN is known as a sex steroid–responsible gene that may be involved in masculinization of the perinatal rat brain (Suzuki et al, 1998, 2009; Suzuki & Nishiahara, 2002; Chiba et al, 2007). In adult rats, PGRN gene expression is up-regulated by estrogen in the hippocampus and PGRN may mediate the mitogenic effects of estrogen in the active area of neurogenesis. PGRN deficiency has been shown to affect sexual behavior and anxiety in mice in a male-specific manner, possibly through reducing the levels of serotonergic receptor 5-HT1A (Kayasuga et al, 2007; Petkau et al, 2012). In addition, females have been reported to have increased Aβ load compared with males (Oakley et al, 2006; Bundy et al, 2019).

It should be noted that APP and Aβ levels were not altered by PGRN loss in the 6-mo-old male APPswe/PS1ΔE9 (APP/PS1) mice (Takahashi et al, 2017). Another study found that PGRN deficiency does not affect APP levels and processing in cortical lysates of 9–13-mo-old PDGF-APP$_{Sw,Ind}$ J9 (APP$^{low}$) mice, although mice of mixed sex were used (Minami et al, 2014). Although causes for the differences in these studies remain to be determined, it is possible that PGRN only affects the metabolism of specific APP variants at a certain age. PGRN might interacts with factors involved in APP trafficking, processing and turnover in a sex- and age-dependent manner to regulate the levels of specific APP variants in neurons.

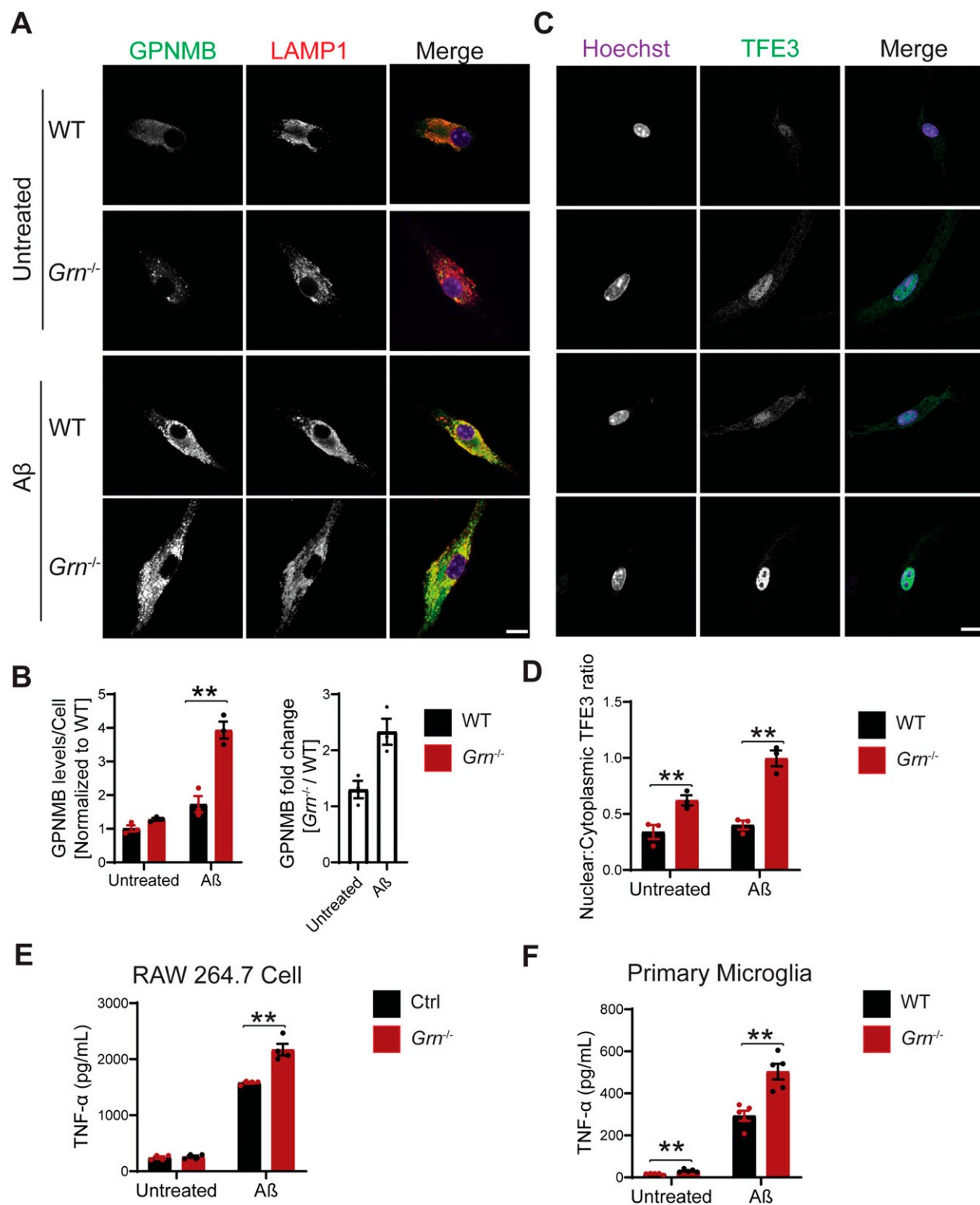

**Figure 12. Aβ treatment leads to enhanced inflammation and lysosomal responses in PGRN-deficient microglia.**
**(A)** Representative confocal images of primary microglia treated with 10 μM Aβ fibrils for 24 h and stained with anti–glycoprotein nonmetastatic melanoma protein B (GPNMB) and anti-LAMP1 antibodies. Scale bar = 10 μm. **(A, B)** Quantification of GPNMB levels per cell for experiment in (A). Mean ± SEM; n = 3, two-way ANOVA, **P < 0.01.
**(A)** Quantification of GPNMB fold changes ($Grn^{-/-}$/WT) for experiment in (A): t test, *P < 0.05. **(C)** Immunostaining of TFE3 in WT and $Grn^{-/-}$ primary microglia treated with and without Aβ fibrils. Scale bar = 10 μm. **(C, D)** The ratio of nuclear to cytoplasmic TFE3 signals were quantified for experiment in (C). Mean ± SEM; n = 3, two-way ANOVA, **P < 0.01. **(E)** Raw 264.7 cells were either untreated or treated with 10 μM Aβ fibrils for 24 h and TNF-α levels in the medium were quantified. n = 4, two-way ANOVA, **P < 0.01.
**(F)** Primary microglia were either untreated or treated with 10 μM Aβ fibrils for 24 h and TNF-α levels in the medium were quantified. n = 5, two-way ANOVA, **P < 0.01.

In addition to AD, *GRN* haplotypes may influence the risk of other neurodegenerative disease like primary progressive multiple sclerosis in a gender-specific manner (Fenoglio et al, 2010). Gender related risk factors might modulate PGRN-mediated phenotypic expression (Piscopo et al, 2013; Curtis et al, 2017).

Because of the drastic effect of PGRN on Aβ pathology in males, we have analyzed microglial responses to Aβ plaques in female and male mice separately, but we did not detect any sex difference in microglial responses to Aβ plaques (Figs 4F and 9B). In both males and females, PGRN deficiency leads to an up-regulation of lysosomal enzyme cathepsins, lysosome membrane protein CD68, and the cell surface receptor TREM2, as well as GPNMB, galectin-3 specifically in microglia around Aβ plaques (Figs 4F and 9B and data not shown). Furthermore, our data show that PGRN deficiency in primary microglia results in impairment of lysosome membrane integrity and increased lysosomal and inflammatory responses after Aβ fibrils treatment. Our results support that PGRN regulates microglia-mediated inflammation through modulating lysosome homeostasis.

Despite increased microglial inflammation in the absence of PGRN, PGRN loss does not seem to have a major effect on Aβ levels in the 10-mo-old mice, although a reduction in the number and area of Aβ plaques was observed (Fig 3), indicating that Aβ concentration might be higher at each plaque in PGRN-deficient mice. It is possible that microglia internalize Aβ but not necessarily degrade Aβ because of poor lysosomal acidification (Sole-Domenech et al, 2016). In addition, PGRN deficiency has been shown to increase microglia-mediated synaptic pruning (Lui et al, 2016), which could have detrimental effects in AD mouse models. Thus the effect of PGRN on cognitive function in AD is confounded by multiple factors.

## Lysosomal abnormalities and enhanced inflammation in PGRN-deficient mice

Our data support that PGRN help maintain proper lysosome homeostasis in microglia upon Aβ stimulation. First, we found that PGRN is highly expressed in microglia near Aβ plaques (Fig S1A and B). Second, PGRN deficiency leads to increased expression of lysosome proteins near the Aβ plaques in vivo (Figs 8A–C and 9A–C). Furthermore, we found that lysosome abnormalities caused by PGRN deficiency are correlated with an increase in several genes associated with microglial activation, including CD68, galectin-3, TREM2, and GPNMB, which are up-regulated specifically in the microglia near the plaque (Figs 4–6). These genes are part of the DAM signature initially identified in 5XFAD mouse model, however, their characteristics were since further validated in other AD mouse models, such as the PS2APP and APP/PS1 (Kamphuis et al, 2016; Deczkowska et al, 2018; Friedman et al, 2018; Mrdjen et al, 2018). Accumulating evidence shows that the DAM phenotype is a common signature of microglial response to central nervous system (CNS) pathology, irrespective of the disease etiology (Deczkowska et al, 2018). However, recent studies found human microglia have an AD-related gene signature distinct from the DAM signature seen in mouse models (Zhou et al, 2020). Despite the differences, the researchers also noted some commonalities, including many genes involved in lipid and lysosomal biology (Del-Aguila et al, 2019; Masuda et al, 2019; Mathys et al, 2019; Srinivasan et al, 2020).

We have found an up-regulation of galectin-3, a β-galactoside–binding cytosolic lectin, in PGRN-deficient microglia in response to Aβ fibril in vivo and in vitro (Figs 4 and 11). In addition, we demonstrated that PGRN-deficient primary microglia are more susceptible to lysosome damage (Fig 10). Galectin-3 has been shown to get recruited to lysosomes upon lysosome membrane permealization (Aits et al, 2015; Chauhan et al, 2016). On the lysosome, Galectin-3 unifies and coordinates endosomal sorting complexes required for transport (ESCRT) and autophagy responses to lysosomal damage (Jia et al, 2020). This could suggest that PGRN loss results in increased lysosome damage and enhanced galectin-3 dependent responses attempting to repair lysosomes or clear damaged lysosomes in response to Aβ stimulation. The mechanism involved in galectin-3 up-regulation is still unclear. TREM2 has been shown to be required for transcriptional activation of a set of DAM genes. However, galectin-3 is not among the list (Kamphuis et al, 2016; Deczkowska et al, 2018; Friedman et al, 2018; Mrdjen et al, 2018).

Another DAM marker up-regulated in the microglia near the Aβ plaque in the PGRN-deficient mice is GPNMB, a type I transmembrane protein that contains an N-terminal signal peptide, an integrin-binding (RGD) motif and a polycystic kidney disease domain in its extracellular domain (ECD), a single pass transmembrane domain and a short cytoplasmic tail harboring a lysosomal/endosomal sorting signal (Maric et al, 2013). GPNMB was first discovered in the bone of an osteopetrotic rat model (Safadi et al, 2001). Aside from its anabolic function in the bone, emerging evidence suggests that GPNMB has anti-inflammatory and reparative functions (Budge et al, 2018). GPNMB has also been demonstrated to be neuroprotective in an animal model of amyotrophic lateral sclerosis, cerebral ischemia, and other disease models (Srinivasan et al, 2016; Budge et al, 2018). The expression of GPNMB is up-regulated in several lysosomal storage diseases and is correlated with lipid accumulation in the lysosome (van der Lienden et al, 2018). In terms of neurodegenerative diseases, the levels of GPNMB are elevated in brain samples of sporadic AD patients and in the substantia nigra of sporadic Parkinson's disease patient brains (Hüttenrauch et al, 2018; Moloney et al, 2018). We found that GPNMB is specifically up-regulated in *Grn*[−/−] microglia near the Aβ plaques and overlaps with galectin-3 in the microglia (Fig 5E–G). Although the function of GPNMB in microglial responses remains to be determined, our results indicate a connection between PGRN deficiency and GPNMB function.

It should be noted that a recent proteomic study has found that galectin-3 and GPNMB are among the two most up-regulated proteins in *Grn*[−/−] mouse brain and FTLD patients with *GRN* mutations and the authors have proposed that GPNMB could be used as a specific marker for FTLD-*GRN* (Huang et al, 2020). Our results are consistent with these findings and suggest a connection between galectin-3, GPNMB, and PGRN function at the lysosome.

## Crosstalk between lysosomal abnormalities and inflammatory responses

Microglia are tissue-resident immune cells of the CNS, which constantly survey their microenvironment with their processes. They regulate immune homeostasis in the healthy and degenerating CNS. Microglia detect damage-associated molecular patterns

from dying neurons and chronic microglia activation is a hallmark shared in many neurodegenerative diseases (Hanisch & Kettenmann, 2007; Heneka et al, 2014).

However, how lysosome abnormalities contribute to microglia activation is still unclear. In this study, we examined the connection between lysosome abnormality and microglia activation in PGRN-deficient conditions. First, we showed that PGRN-deficient microglia are more susceptible to LMP upon treatment with Aβ fibrils and LLOME. LMP has been shown to result in the release of lysosomal enzymes into the cytosol, which often leads to inflammasome activation (Weber & Schilling, 2014; Katsnelson et al, 2016; Nebel et al, 2017). Thus, lysosome membrane integrity is closely linked to inflammation. Furthermore, we demonstrated that PGRN deficiency causes increased inflammatory responses upon Aβ fibril treatment. In vivo, PGRN deficiency leads to enhanced microglia activation near the Aβ plaques (Figs 4–6). In vitro, deletion of PGRN in microglia results in higher level of TNF-α upon Aβ fibril treatment (Fig 12E and F). We have found increased levels of TFE3 in the PGRN-deficient microglia in response to Aβ both in vivo and in vitro and activation of TFE3/TFEB has been shown to drive the expression of not only lysosomal genes but also inflammation genes in several studies (Nabar & Kehrl, 2017; Brady et al, 2018). Unfortunately, we cannot determine the levels of TFEB and MiTF in PGRN-deficient microglia because of unavailability of specific antibodies suitable for tissue section staining. Another transcription factor that might connect lysosomal dysfunction to inflammation is Stat3. It has been that substrate accumulation and lysosome stress can activate Stat3, which is known to mediate the transcription of many genes involved in inflammatory responses (Martinez-Fabregas et al, 2018).

In summary, we show that PGRN deficiency results in a reduction in the levels of mutant human APP in males but not females at the young age. PGRN deficiency leads to enhanced lysosome abnor-malities and inflammatory responses in response to Aβ in vivo and in vitro. These studies provide novel insights into understanding of neurodegenerative diseases associated with PGRN haploinsufficiency or polymorphisms, including FTLD and AD.

# Materials and Methods

## Primary antibodies and reagents

The following antibodies were used in this study: Mouse anti-β-Amyloid (SIG-39320; BioLegend), rabbit anti-β-Amyloid (25524-1-AP; Proteintech Group), mouse anti-galectin-3 (126702; BioLegend), goat anti–galectin-3 (AF1154; R&D Systems), goat anti-GPNMB (AF2330; R&D Systems), mouse anti-GAPDH (60004-1-Ig; Proteintech Group), rat anti-mouse LAMP1 (553792; BD Biosciences), LAMP2 (Developmental Studies Hybridoma Bank, GL2A7-c), goat anti-CathB (AF965; R&D Systems), goat anti-CathD (AF1029; R&D Systems), goat anti-CathL (AF1515; R&D Sys-tems), sheep anti-TREM2 (AF1729; R&D Systems), rabbit anti IBA-1 (01919741; Wako), goat anti-AIF-1/Iba1 (NB100-1028; Novus Biologicals), rat anti-CD68 (MCA1957; Bio-Rad), rabbit anti-TFE3 (HPA023881; Sigma-Aldrich), and sheep anti-PGRN (AF2557; R&D Systems). PS1-related peptides were detected using Ab14, a rabbit polyclonal antibody gen-erated against amino acids 1–25 of PS1. Rabbit anti-BACE1 antibody were

raised against the sequence CLRQQHDDFADDISLLK from 485 to 501 of BACE1 protein (Yan et al, 2001).

The following reagents were also used in the study: DMEM (10-017-CV; Cellgro), HBSS (21-020-CV; Cellgro), DMEM/Ham's F-12 (DMEM/F-12) (10-092-CV; Cellgro), 0.25% Trypsin (25-053-CI; Corning), β-Amyloid (1–42) (AS-60883; Anaspec), Thioflavin S (T1892; Sigma-Aldrich), TrueBlack Lipofuscin Autofluorescence Quencher (23007; Biotium), L-Leucyl-L-Leucine methyl ester (LLOME) (16008; Cay-man), Odyssey blocking buffer (927–40000; LI-COR Biosciences), protease inhibitor (05056489001; Roche), Pierce BCA Protein Assay Kit (23225; Thermo Fisher Scientific), OCT compound (62550-01; Electron Microscopy Sciences), and mouse TNF-α ELISA Kit (430901; BioLegend).

## Mouse strains

C57/BL6, Grn$^{−/−}$ (Yin et al, 2010), and 5XFAD (Oakley et al, 2006) mice were obtained from the Jackson Laboratory. Both male and female mice were used and analyzed in separate groups. All the mice were housed in the Weill Hall animal facility at Cornell. All animal procedures have been approved by the Institutional Animal Care and Use Committee at Cornell.

## Behavioral test

4.5-mo-old male and female WT, Grn$^{−/−}$, 5XFAD and 5XFAD Grn$^{−/−}$ mice in the C57/BL6 background (6–12 mice/group) were subject to the following behavioral tests: (1) Open-field test: Mice were placed in a clear plastic chamber (30 × 30 × 30 cm) for 5 min. The total track distance, center track distance, center time, and center entries were tracked by the Viewer III software (Biobserve). The apparatus was thoroughly cleaned with 70% ethanol between trials. (2) Y-maze test: Spatial working memory performance was assessed by re-cording spontaneous alternation behavior in a Y-maze. The Y-maze was made of light grey plastic and consisted of three arms at 120°. Each arm was 6-cm wide and 36-cm long and had 12.5-cm high walls. Each mouse was placed in the Y-maze and allowed to move freely during an 8 min session. The series of arm entries was recorded visually and arm entry was considered to be completed when the hind paws of the mouse were completely placed in the arm. The maze was cleaned with 70% ethanol after each mouse. Alternation was defined as successive entries into the three arms on overlapping triplet sets (e.g., ABC and BCA). The percentage alternation was calculated as the ratio of actual to possible alternations (defined as the total number of arm entries minus two). For all behavioral an-alyses, experimenters were blind to the genotypes of the mice.

## Cell culture and biochemical assays

Raw 264.7 cells were maintained in DMEM (Cellgro) supplemented with 10% fetal bovine serum (Sigma-Aldrich) in a humidified incu-bator at 37°C with 5% CO$_2$. Raw 264.7 cells with PGRN deletion or controls were generated by infecting the cells with lentivirus expressing Cas9 and guide RNAs (GCTCCCTGGGAGGCATCTGG and CGGACCCCGACGCAGGTAGG) targeted to mouse PGRN exon 1 (oligos with 5'-cacc gGCTCCCTGGGAGGCATCTGG-3' and 5'-aaac CCAGATG-CCTCCCAGGGAGCc-3' or 5'-cacc gCGGACCCCGACGCAGGTAGG-3' and

5′-aaac CCTACCTGCGTCGGGGTCCGc-3′ were ligated to pLenti-CRISPRv2 [Addgene]) or Cas9 only. Cells were selected with puromycin (8 μg/ml) 2 d after infection and the knockout is confirmed by Western blot and immunostaining.

WT and Grn−/− mouse primary microglia were isolated from P0 to P2 pups and grown on astrocytes for 14 d before being shaken off according to a published protocol (Zhou et al, 2017c).

For LLOME treatment assays, 500 μM LLOME (Cayman) was added to Raw 264.7 cells and primary microglia for 2 h before fixation.

### Tissue preparation for Western blot analysis

Mice were perfused with PBS and tissues were dissected and snap-frozen with liquid nitrogen and kept at –80°C. On the day of the experiment, frozen tissues were thawed and homogenized on ice with bead homogenizer (Moni International) in ice-cold radioimmunoprecipitation assay buffer (RIPA) (150 mM NaCl, 50 mM Tris–HCl [pH 8.0], 1% Triton X-100, 0.5% sodium deoxycholate, and 0.1% SDS) with 1 mM PMSF, and 1× protease inhibitors (Roche). After centrifugation at 14,000g for 15 min at 4°C, supernatants were collected as the RIPA-soluble fraction (soluble fraction). The insoluble pellets were washed with RIPA buffer and extracted in guanidine buffer (5 M guanidine, 50 mM Tris–HCl [pH 8.0], and 1× protease inhibitors). After all pellets were dissolved completely, samples were centrifuged at 20,000g for 15 min at 24°C and the supernatants were collected as the guanidine-soluble fraction (insoluble fraction). Protein concentrations were determined via BCA assay, and then standardized. Samples were run on 12% or 15% polyacrylamide gels, and then transferred to Immobilon-FL polyvinylidene fluoride membranes (Millipore Corporation). Membranes were blocked with either 5% non-fat milk in PBS or Odyssey Blocking Buffer (LI-COR Biosciences) for 1 h and then incubated with primary antibodies, rocking overnight at 4°C. Membranes were then washed with Tris-buffered saline with 0.1% Tween-20 (TBST) three times, 10 min each, and incubated with fluorescently tagged secondary antibodies (LI-COR Biosciences) for 1 h at room temperature, followed by three washes. Membranes were scanned using an Odyssey Infrared Imaging System (LI-COR Biosciences). Densitometry was performed using Image Studio (LI-COR Biosciences) and ImageJ.

### Thioflavin S staining

Thioflavin S staining was performed as previously reported (Ly et al, 2011). 20-μm-thick mice brain cryosections were stained with Thioflavin S solution (1%, in 80% Ethanol) for 15 min, followed by washing with 80% ethanol, 70% ethanol, and water. For double and triple staining with antibodies, after Thioflavin S staining, sections were incubated with primary antibodies overnight at 4°C. The next day, sections were incubated with secondary antibodies (Hoechst) at room temperature for 1 h and mounted onto cover slips. Images were acquired using a 20× objective on a Leica DMi8 inverted microscope. Images were captured from the cerebral cortex and hippocampus regions and images from ≥5 brains per genotype were used for quantitative analysis.

### Immunofluorescence staining, image acquisition, and analysis

Cells were fixed, permeabilized with 0.05% saponin, and visualized using immunofluorescence microscopy as previously described (Brady et al, 2013). For brain section staining, mice were perfused with cold PBS and tissues were post-fixed with 4% paraformaldehyde. After dehydration in 30% sucrose buffer, tissues were embedded in OCT compound (Electron Microscopy Sciences). 20-μm-thick brain sections were cut with cryotome. Tissue sections were blocked and permeabilized with 0.1% saponin in Odyssey blocking buffer or 0.2% Trion X-100 in Odyssey blocking buffer before incubating with primary antibodies overnight at 4°C. The next day, sections were washed 3× with cold PBS followed by incubation with secondary fluorescent antibodies and Hoechst at room temperature for 1 h. The slides were then mounted using mounting medium (Vector laboratories). To block the autofluorescence, all sections were incubated with 1× TrueBlack Lipofuscin Autofluorescence Quencher (Biotium) in 70% ethanol for 30 s at room temperature after the staining process. Antigen retrieval was performed by microwaving in sodium citrate buffer (pH 6.0) for 10 min. Images were acquired on a CSU-X spinning disc confocal microscope (Intelligent Imaging Innovations) with an HQ2 CCD camera (Photometrics) using 40× and 100× objectives, 8–10 different random images were captured. Lower magnification images were captured by 10× or 20× objectives on a Leica DMi8 inverted microscope, three to five images were captured from each sample. Data from ≥3 brains in each genotype were used for quantitative analysis.

To quantify ThioS-positive plaques and Aβ plaques, plaque area, plaque number and intensity were calculated using the "analyze particles" function of ImageJ. For the quantitative analysis of IBA1 and CD68 levels in the brain sections, the fluorescence intensity and area were measured directly using ImageJ after a threshold application. The protein levels were determined by the total fluorescence signals. For the quantitative analysis of IBA1 levels near Aβ plaques, the Aβ plaques were selected using the region of interest (ROI) tool after the data channels were separated (Image\Color\Split Channels). Next, all IBA1-positive microglia that fully or partially overlap with the Aβ plaques signal were selected (Analyze\tools\ROI manager), Subsequently, the fluorescence area and intensity were measured. For CD68, galectin-3, GPNMB, TREM2, cathepsin B (CathB), and cathepsin D (CathD) levels near Aβ plaques, the images were analyzed using the same procedure in ImageJ. To quantify nuclear TFE3 levels in the microglia, the nucleus was selected based on Hoechst staining and the fluorescence intensity was measured directly by ImageJ. Three brain sections per mouse, separated by 100 μm, were used for quantification. The mean from the three sections was used to be representative of each mouse. Data were normalized to age-matched 5XFAD controls.

For the quantitative analysis of intracellular levels of galectin-3, CathD, and GPNMB in primary microglia culture, the entire cell body was selected, and the fluorescence intensity was measured directly using ImageJ after a threshold application. To quantify the degree of colocalization between galectin-3 and the lysosomal marker LAMP1, the JACoP plugin was used to generate Manders' coefficients. To quantify nuclear to cytoplasmic TFE3 ratio, the nucleus and the

entire cell were selected, and the fluorescence intensity was measured directly by ImageJ. Intensity from the entire cell were subtracted from nuclear to calculate cytoplasmic TFE3 intensity. For each experiment, at least 12 pairs of cells were measured and the data from three to five independent experiments were used for statistical analysis.

### Preparation of Aβ fibrils

Aβ fibrils were prepared as described previously with slight modification (Solé-Domènech et al, 2018). One milligram of Aβ (Anaspec) was solubilized at 1 mg/ml in 1 ml of 50 mM sodium tetraborate, pH 9.3. To form fibrils, Aβ was immediately diluted in pH 7.4 PBS buffer (50 $\mu$M). The fibrillation mixture was prepared in sealed polycarbonate ultracentrifuge tubes and incubated for 24 h at 37°C under constant rotation. After aggregation, centrifuge the mixture at 15,000$g$ for 30 min and carefully discard the supernatant without disturbing the pellet, the pellets were resuspended in pH 7.4 PBS buffer (125 $\mu$M). Aβ fibrils were then added to cells at the indicated concentrations for 24 h and cells were collected for subsequent experiments.

### Enzyme-linked immunosorbent assay

Aβ42 peptides in cortical lysates were captured by Aβ$_{33-42}$ specific antibody (21F12). Aβ40 peptides in cortical lysates were captured by Aβ$_{40}$ specific antibody (2G3). Both captured Aβ42 and Aβ40 were detected by biotinylated mouse anti–Aβ$_{1-16}$ antibody (6E10) followed by streptavidin HRP (The Jacksonimmuno). All ELISAs were developed using ELISA TMB (Sigma-Aldrich) and absorbance read on a Bio-Tek plate reader. The synthetic human Aβ$_{1-42}$ or Aβ$_{40}$ peptide (AnaSpec) was used for standard curves for each assay. To determine the inflammatory responses, WT, $Grn^{-/-}$ primary microglia and control, $Grn^{-/-}$ Raw 264.7 cells were treated with 10 μM Aβ fibrils for 24 h. To measure TNF-$\alpha$ levels, conditioned medium was collected and analyzed using mouse TNF-$\alpha$ ELISA Kit (BioLegend) according to the manufacturer's instruction.

### RNA-seq analysis

RNAs were extracted from brain hippocampus region using Trizol (Thermo Fisher Scientific). RNA quality was checked using Nano-Drop, gel electrophoresis and Agilent 4200 TapeStation. cDNA library was then generated using the QuantSeq 3′ mRNA-Seq Library Prep Kit FWD for Illumina (Lexogen). 86 bp single-end sequencing was performed on an Illumina NextSeq500 (Illumina) using services provided by the Cornell Biotech Facilities. Data quality was assessed using FastQC (https://www.bioinformatics.babraham.ac.uk/projects/fastqc/). Reads that passed quality control were aligned to reference genome (Ensembl 98.38) (Zerbino et al, 2018) using STAR (Dobin et al, 2013). The number of reads in each gene was obtained using HTSeq-count (Anders et al, 2015). Differential expression analysis was performed using the R package edgeR (Robinson et al, 2010), followed by the limma package with its voom method (Law et al, 2014). Genes with FDR control $P$-value ≤ 0.05 and log fold change ≥ 0.5 were identified as differentially expressed genes. Heat maps were made using the R package gplots.

### Statistical analysis

All statistical analyses were performed using GraphPad Prism 8. The normality of data was tested by the Shapiro–Wilk test ($P > 0.05$). All data are presented as mean ± SEM. Statistical significance was assessed by unpaired $t$ test (for two groups comparison), one-way ANOVA tests or two-way ANOVA tests with Bonferroni's multiple comparisons (for multiple comparisons). $P$-values less than or equal to 0.05 were considered statistically significant. *$P$ < 0.05; **$P$ < 0.01.

## Data Availability

The data supporting the findings of this study are included in the Supplemental Data 1. Additional data are available from the corresponding author on request. No data are deposited in databases.

## Supplementary Information

## Acknowledgements

We would like to thank Xiaochun Wu for technical assistance, Dr Tony Bretscher's lab for assistance with confocal microscope, Dr Tobias Dörr's lab for assistance with Leica DMi8 inverted microscope, Dr. Fred Maxfield's lab for assistance with Aβ fibril preparation, Dr Thomas Cleland's lab, and Dr Chris Schaffer and Dr Nozomi Nishimura's lab for assistance with behavioral tests. We would like to thank Dr Steven M. Paul and Dr Ronald Demattos from Eli Lilly for providing 2G3 and 21F12 antibodies. This work is supported by NINDS/NIA (R01NS088448 and R01NS095954) and the Bluefield project to cure frontotemporal dementia to F Hu, and NIA (R01AG064239) to W Luo.

### Author Contributions

H Du: data curation, formal analysis, validation, visualization, methodology, and writing—original draft, review, and editing.
MY Wong: data curation and formal analysis.
T Zhang: data curation and formal analysis.
MN Santos: data curation and formal analysis.
C Hsu: data curation and writing—review and editing.
J Zhang: resources, software, and formal analysis.
H Yu: resources, software, and supervision.
W Luo: formal analysis, supervision, methodology, project administration, and writing—review and editing.
F Hu: conceptualization, resources, formal analysis, supervision, funding acquisition, investigation, project administration, and writing—original draft, review, and editing.

## Conflict of Interest Statement

The authors declare that they have no conflict of interest.

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
