## [Reviewer comments · Life Science Alliance]

Life Science Alliance

A multifaceted role of progranulin in regulating amyloid-beta dynamics and responses

Huan Du, Man Wong, Tingting Zhang, Mariela Santos, Charlene Hsu, Junke Zhang, Haiyuan Yu, Wenjie Luo, and Fenghua Hu

DOI: <https://doi.org/10.26508/lsa.202000874>

Corresponding author(s): Fenghua Hu, Cornell University

Review Timeline:

Submission Date:	2020-08-12
Editorial Decision:	2020-09-11
Appeal Received:	2021-02-02
Editorial Decision:	2021-02-05
Revision Received:	2021-02-05
Editorial Decision:	2021-03-26
Revision Received:	2021-03-31
Editorial Decision:	2021-05-20
Revision Received:	2021-05-26
Accepted:	2021-05-26

Scientific Editor: Shachi Bhatt

Transaction Report:

September 11, 2020

Re: Life Science Alliance manuscript #LSA-2020-00874-T

Dr. Fenghua Hu
Cornell University
Weill Institute for Cell and Molecular Biology and Department of Molecular Biology and Genetics
345 Weill Hall, Cornell University
Ithaca, NY 14853

Dear Dr. Hu,

Thank you for submitting your manuscript entitled "Lysosome abnormalities caused by progranulin deficiency lead to enhanced inflammatory responses to amyloid-beta plaques". The manuscript has been evaluated by expert reviewers, whose reports are appended below. Although we and the reviewers are intrigued by the findings, we are concerned that the revisions required by the referees are more substantial than can be addressed in a typical revision period.

Given the interest in the topic, I would be open to resubmission to Life Science Alliance of a significantly revised and extended manuscript that addresses most of the reviewers' concerns and is subject to further peer-review, preferably by the same set of referees. The concern about conceptual advance (Rev 1) and behavioral analysis (Rev 1 and 2) and figuring out the mechanism of sex differences (Rev 1 and 3) do not need to be addressed for a publication in Life Science Alliance. All other concerns, particularly the ones pertaining to the effects of progranulin deficiency on the cellular function of microglia and quantification of Abeta pathology, and others should be addressed for the manuscript to be reconsidered at Life Science Alliance.

To resubmit a significantly revised manuscript to Life Science Alliance you may submit an appeal directly through our manuscript submission system, along with a point-by-point rebuttal. Please note that revised manuscript will undergo an editorial evaluation for reassessment of priority and novelty prior to being sent back to the referees.

Regardless of how you choose to proceed, we hope that the comments below will prove constructive as your work progresses. We would be happy to discuss the reviewer comments further once you've had a chance to consider the points raised in this letter.

Thank you for thinking of Life Science Alliance as an appropriate place to publish your work.

Sincerely,

Shachi Bhatt, Ph.D.
Executive Editor
Life Science Alliance

Reviewer #1 (Comments to the Authors (Required)):

In the present study, Du et al. investigate the role of PGRN in microglial responses to Abeta plaques using 5XFAD mice and primary microglial culture. The results show that PGRN is upregulated in microglia near Abeta plaques. They also report that PGRN deficiency leads to a decrease in Abeta plaques in male 5XFAD mice at 4.5 months of age, enhanced microgliosis, and an increase in expression of some lysosomal and inflammatory proteins such as cathepsin B and TREM2 in microglia near Abeta plaques. In vitro studies using microglia culture support some results from their mouse studies.

Unfortunately, nearly all of the data obtained from the present study are already reported in very similar studies, or can be easily predicted from the previous publications. Numerous studies have reported on Grn^{-/-} crossed with transgenic AD mice, and it is not clear that the current study substantially advances the field. Largely, the current study is confirmatory, adding to the list of lysosomal marker proteins reported in this context.

Below are specific comments.

1. NOVELTY: Most importantly, the novelty of this manuscript is significantly compromised by previous publications. The relationship between increased lysosomal biogenesis and microglial activation has been investigated in TBI models and very similar results have been obtained (Tanaka et al, 2013, Neuroscience). PGRN is well known to be increased near Abeta plaques in AD mouse models and human AD (Baker et al, 2006, Nature; Pereson et al, 2009, J Pathol; Minami et al, 2014, Nat Med; Gowrishankar et al, 2015, PNAS; Mendsaikhan et al, 2019, Acta Neuropathol Commun). Reduced Abeta burden and increased microglial response near Abeta plaques have been already reported in previous studies (Takahashi et al, 2017, Acta Neuropathol; Hosokawa et al, 2018 Exp Anim; Götzl et al, 2019, EMBO Mol Med). A previous study has shown partial DAM signature in Grn^{-/-} microglia (Götzl et al, 2019, EMBO Mol Med).

2. CONTEXT: In the Introduction, the background of PGRN is described in detail, but its role Alzheimer's disease is completely lacking. Numerous previous studies investigating the role of PGRN in AD are not reviewed. It is essential that the Introduction and to some extent the Abstract make clear the extensive existing information, and indicate what new question is solved by repeating the previous work with a different AD strain. The literature cannot be ignored until the reader reaches the Discussion.

3. FUNCTION: The paper documents in detail altered protein level and distribution for many lysosomal constituents in the Grn deficient 5XFAD condition. However, there is essentially no study of function. For the mice, Abeta plaque is decreased in males and unchanged in females. No change in behavior assessment of the 5XFAD mice is detected. At the cellular level, no assays of cell function such as phagocytosis or lysosomal degradation for Abeta or other molecules are assayed. The few pieces of information in the manuscript suggest no major change in mouse or cell function. The focus is specifically on changes in the expression of numerous lysosomal markers. Any statement that this manuscript documents lysosomal "dysfunction" needs to be deleted from Abstract, Introduction, Results and Discussion.

4. SEX: The finding of a male-specific change in Abeta plaques by PGRN deficiency is intriguing, but this observation is simply reported, and then not studied at all. Whether Abeta plaques are changed by PGRN loss at older ages (e.g., 10 m) in male and female remains unclear. Did the authors use primary microglia from female versus male mice? What is molecular basis for this difference?

5. In DAM state, microglia express hundreds of DAM signature genes (Keren-Shaul et al, 2017, Cell). Only a few were shown to be increased in the present study and more typical DAM genes are not shown to be upregulated. It is thus unclear whether PGRN deficiency induces DAM state or another undefined state in microglia near Ab plaques. The authors should tone down and avoid stating that PGRN loss results in DAM gene expression.

6. In transcriptomic analysis of the present study, DEGs in 4.5m GRN^{-/-} and 5XFAD vs WT have not been detected, which seems inconsistent with previous studies and the fact that there exist a plenty of amyloid plaques in 5XFAD at this age. It might have something to do with FC threshold of the analysis. But because of this discrepancy, transcriptomic changes in 4 genotypes are not convincing and requires re-analysis and/or detailed explanation.

7. In Figure 8D and 8E, the result of TFE3 translocation is less than convincing. It looks as though the PGRN deficiency causes just an increase, but not nuclear translocation, of TFE3 under the current magnification. The authors should use higher magnification for this analysis and nuclear/cytoplasmic TFE3 ratio should be calculated as shown in in vitro studies in Figure 11D. In addition, the correlation with an increase in galectin-3 should be quantitatively assessed.

8. It is stated that "PGRN deficiency in microglia results in impairment of lysosomal membrane integrity and increased inflammatory responses after phagocytosis of Abeta fibrils". However, the statement is not supported by the authors' results. First, phagocytosis of fibrils has not been assessed in this study. It is unlikely that the fibrils without sonication are properly phagocytosed. Second, there are no data showing Abeta induces impairment of lysosomal membrane integrity. It looks like Abeta just increases galectin-3 and the colocalization analysis shown in Figure 9 has not been performed.

9. In Figure 9, quantification of untreated groups is missing. It is thus unclear whether LLOME is effective in Ctrl/WT.

10. In all bar graphs, data points should be presented using dot plots to show the distribution. In addition, the authors should describe how normality was estimated in the method.

11. In Figure 3B, 10 and 11, a t-test is inappropriately used for the statistical analysis. The authors should use one-way or two-way ANOVA for the results.

Minor comments:

12. It would be helpful if the various panels and graphs were labeled with age and sex of animals and the brain region analyzed.

13. In Figure 8D and 11C, while Dapi appeared to be used for nuclei staining, the method says Hoechst was used.

14. In Figure 11D, it looks like the data are normalized to Grn^{-/-} treated with Abeta but it is not described anywhere.

Reviewer #2 (Comments to the Authors (Required)):

In this manuscript, the authors report that PGRN deficient decreases Abeta plaques in male, not female, 5XFAD mice. Further, the authors claim that PGRN regulates microglia-mediated inflammation through modulating lysosomal function in response to Amyloid beta plaques. The paper is well written and easy to understand. However, some critical primary data is not carefully performed making the overall interpretation of the data uncertain. These major concerns need to be addressed to make this paper suitable for publication.

1) The most major concern are the experiments quantifying Abeta plaque pathology in Fig. 2. In general, they are poorly done and Immunofluorescence is not the best approach for quantifying Amyloid plaque burden. Also, 5X FAD mice have a substantial amount of variability in plaque number and size at 4.5 months of age. Much larger numbers of mice per group are necessary to draw any solid conclusion. General experimental details are missing. For example, how many sections per mice were measured? How far apart were the slices?

2) Further, the 6E10 staining simply does not look right. IHC with antigen retrieval (Formic Acid or PAF) needs to be performed. Finally, a complementary approach is necessary to draw definitive conclusions. Specifically, the inclusion of ELISA measurements of soluble/insoluble Abeta species would allow a precise quantitative measurement.

3) Considering that most of the conclusions are based on immunofluorescence assays, it is essential to include more negative controls for them (i.e. secondary antibodies alone to see how much of the signal could be background). For instance, Fig 2, panel A shows the staining of 6E10 in brain tissue, and the background is very high and the morphology of the "plaque" doesn't look normal. Another example is in Fig 1, panel A, B, and C with PGRN and 6E10 antibodies. This control is important because one of the main conclusions of this work is that PGRN and microglia are recruited to Abeta plaques. Other antibodies with questionable staining are TREM2 (Fig5-G), galectin 3 (fig4-E), CathB (Fig7-C).

4) On page 4 the authors claim that Abeta plaque accumulation decreases the efficiency of PGRN processing based only on a western blot in figure 1 panel D, where they compare levels of full PGRN and GRNs in WT and 5xFAD GRN^{-/-}. It could be interesting to see if treatment with Abeta can impair PGRN processing to make this claim stronger.

5) What antibody was used to detect GRNs? Does the antibody recognize all the different GRN peptides?

6) In figure 3 panel C, it is not clear what IBA1 levels near Ab plaques mean. Just the fluorescence intensity of cells near Ab plaques? How was this data normalized? This question is because it is possible to see an increase of IBA 1 signal in all the picture, not only near Ab plaques, so the data that is has been reported with this graph is not necessarily indicating the recruitment of IBA1 (microglia) to the plaques as a specific response but just an increased pool of activated microglia. So maybe, results should be normalized by the total number of IBA1+ microglia. Same for CD68, CatD, and CatB

7) In Fig. 5 E, a higher magnification should be included for GPNMB staining to enable visualization. Quantification of co-localization could also be performed. Also, in panel G, a lower magnification for TREM2 to have a broad idea of the expression of this protein in the sample.

8) On page 6, the authors claim that the data support the idea that "PGRN is critical for proper lysosome function in microglia in response to Abeta plaques." However, data only shows an

increase in the levels of lysosomal proteins. What about actual function? How does Abeta affect the activity of lysosomal enzymes, such as CatD and B. This data could help explain how Abeta affects PGRN and lysosome function.

9) In Fig. 8 panel D. It would be helpful to add a merged image of TFE3/DAPI with IBA1 and galectin-3 to visualize the data in a better way. How do authors know which cells are close to Abeta plaques in this figure? How is nuclear translocation normalized?

10) Figure 10 and 11. Considering that there is an increase in the levels of GPNMB, which can be seen in the figure but not in the graph, and CatD in untreated GRN^{-/-} cells, and also an increased translocation of TFE3 to the nucleus, it would be useful to graph only the ratio between WT/GRN^{-/-} cells untreated and treated with Abeta. This comparison would allow readers to see if these changes are produced by the deficiency of PGRN or from the addition of Abeta.

11) The co-localization of PGRN and Abeta has been previously studied and associated with lysosomal proteins and microglia. For instance, Mendsaikhan et al., 2019 [10.1177/1533317509346209](https://doi.org/10.1177/1533317509346209). It would be interesting to discuss more in-depth how this new work contributes to the field compared with the existing published data.

12) Moreover, other papers indicate that deficiency of PGRN has a protective role against Abeta accumulation (Hosokawa et al., 2018. [10.1538/expanim.17-0060](https://doi.org/10.1538/expanim.17-0060); Takahashi et al., 2017 [10.1007/s00401-017-1668-z](https://doi.org/10.1007/s00401-017-1668-z)). Do you think this is also associated with gender? How do these previous publications affect the conclusion of this work?

13) Finally how do you account for the discrepancy between your work and Minami et al, Nature Medicine, which convincingly shows Progranulin protects against amyloid β deposition and toxicity in Alzheimer's disease mouse models?

Reviewer #3 (Comments to the Authors (Required)):

This study by Du et al investigates the role of progranulin in pathogenesis of Alzheimer's Disease. Loss-of-function mutations in the progranulin gene, most of which cause haploinsufficiency, are a major cause of frontotemporal dementia, but a polymorphism with milder effects on progranulin levels increases risk for AD. Prior work on progranulin in AD mouse models has yielded mixed results when AD mouse models are crossed with Grn KO mice.

This study is a useful addition to this literature and adds to the data showing possibly counterintuitive protective effects of progranulin deficiency using the 5X FAD model. The discovery of sex-specific effects of progranulin deficiency, and the careful study of microglia are strengths of this paper, particularly insofar as the data begin to connect progranulin's known lysosomal functions with the changes observed in GrnKO microglia exposed to amyloid beta. However, additional data tying the changes in microglia to amyloid pathology would strengthen this paper. Additional study of sex and age differences in amyloid plaque load and microglial activation in the GrnKO:5XFAD cross might help clarify the association between microglial changes, amyloid plaques, and behavioral deficits.

1.) The sex difference in the effects of Grn KO is very interesting given the known sex differences in AD risk. It therefore seems that focusing solely on female mice beginning in figure 3 leaves

potentially valuable data unexplored. Measuring microglial activation and localization around plaques in male and female mice of different ages might reveal mechanisms underlying the sex difference in GrnKO:5XFAD plaque burden. For example, do male and female GrnKO:5XFAD mice show differences in plaque load and microglial activation at 2 or 3 months? Perhaps the male mice would have greater microglial activation at these early ages that could result in greater clearance of amyloid?

2.) Were any sex differences in microglia observed in GrnKO mice that didn't carry the 5XFAD transgene?

3.) Female GrnKO:5XFAD mice show striking changes in microglia, but no change in amyloid plaque load or behavior versus Grn WT:5XFAD mice. While there is extensive data showing the importance of microglia in AD pathogenesis, what is the data from this study to establish a link between microglial changes due to progranulin KO and amyloid pathology or behavioral deficits?

February 2, 2021

Dear Dr. Bhatt,

The authors of manuscript #LSA-2020-00874-T have requested an appeal. Their comments are below.

Dear Dr. Shachi Bhatt:

I am writing to re-submit the manuscript "A multifaceted role of progranulin in regulating amyloid-beta dynamics and responses" by Huan Du, Man Ying Wong, Tingting Zhang, Mariela Nunez Santos, Charlene Hsu, Junke Zhang, Haiyuan Yu, Wenjie Luo and myself for consideration as a research article in Life Science Alliance.

The GRN gene, encoding the progranulin (PGRN) protein, has been implicated in many neurodegenerative diseases. Total loss of PGRN leads to neuronal ceroid lipofuscinosis, a lysosome storage disorder. Haploinsufficiency of PGRN is a leading cause of frontotemporal lobar degeneration (FTLD). PGRN is also implicated in Alzheimer's disease (AD). PGRN deficiency results in both lysosome dysfunction and enhanced microglia activation. However, how lysosome dysfunction contributes to microglia activation is still unclear. Furthermore, the role of PGRN in regulating A-beta deposition has been controversial in the field. In this manuscript, we report that PGRN deficiency leads to a decrease in A-beta deposition in a sex and age dependent manner. PGRN loss leads to a reduction in the levels of human mutant APP specifically in young male mice. We also provide evidence supporting that lysosome dysfunction caused by PGRN loss contributes to enhanced microglia activation in response to A-beta stimulation in vitro and in vivo. Thus we believe that this manuscript will be of great interest to readers in the FTLD, AD, and lysosome field.

We have added new data and revised our manuscript extensively to address the reviewers' and the editor's concerns. In addition, we have determined the changes in human APP levels to elucidate the mechanism underlying the sex specific effect of PGRN on A-beta. This is a novel finding regarding the role of PGRN in AD and might explain a previous report showing that GRN polymorphisms may influence AD risk in a male-specific manner (Viswanathan, J. et al. 2009). Please see the attached "Responses to reviewers' comments" for details.

Due to the highly competitive nature of this work and existing competitions, we greatly appreciate an expedited review of our manuscript, if possible.

Thank you very much for your consideration.

Sincerely yours,

Fenghua Hu

We thank the editor and reviewers for their constructive comments regarding our manuscript. During the past 5 months, we have collected new data and revised our manuscript extensively to address the reviewers' and the editor's concerns.

Editor's comments:

Given the interest in the topic, I would be open to resubmission to Life Science Alliance of a significantly revised and extended manuscript that addresses most of the reviewers' concerns and is subject to further peer-review, preferably by the same set of referees. The concern about conceptual advance (Rev 1) and behavioral analysis (Rev 1 and 2) and figuring out the mechanism of sex differences (Rev 1 and 3) do not need to be addressed for a publication in Life Science Alliance. All other concerns, particularly the ones pertaining to the effects of progranulin deficiency on the cellular function of microglia and quantification of A-beta pathology, and others should be addressed for the manuscript to be reconsidered at Life Science Alliance.

Reply: We thank the editor for the constructive feedbacks. We have added the following new data in the revised manuscript to address the reviewers' concerns.

1. Quantification of A-beta levels by ELISA in the 4-month-old and 10-month-old male and female cortical lysates (new Fig. 2A and Fig. 3B) to show age and sex dependent regulation of A-beta by PGRN.
2. Quantification of A-beta pathology in the 10-month-old male and female PGRN deficient mice to show an age dependent regulation of A-beta deposition by PGRN (new Fig. 3A).
3. Western blot analysis to show a specific decrease in the levels of mutant human APP in the young male mice in the absence of PGRN (new Fig. 2B, Fig. S3).
4. Analysis of microglial responses near A-beta plaques in the male PGRN deficient mice to show that PGRN deficiency does not result in sex specific microglial responses to A-beta (new Fig. 4F and 9B).
5. New western blot using cathepsin KO samples to show the specificities of the antibodies used in the study (new Fig. S7).

Our new data demonstrates a novel role of PGRN in regulating APP metabolism in a sex and age dependent manner. In addition to these new exciting data, we have revised the text extensively to address each reviewer's comments. Please see detailed responses below.

Reviewer #1 (Comments to the Authors (Required)):

In the present study, Du et al. investigate the role of PGRN in microglial responses to Abeta plaques using 5XFAD mice and primary microglial culture. The results show that PGRN is upregulated in microglia near Abeta plaques. They also report that PGRN deficiency leads to a decrease in Abeta plaques in male 5XFAD mice at 4.5 months of age, enhanced microgliosis, and an increase in expression of some lysosomal and inflammatory proteins such as cathepsin B and TREM2 in microglia near Abeta plaques. In vitro studies using microglia culture support some results from their mouse studies.

Unfortunately, nearly all of the data obtained from the present study are already reported in very similar studies, or can be easily predicted from the previous publications. Numerous studies have reported on Grn^{-/-} crossed with transgenic AD mice, and it is not clear that the current study substantially advances the field. Largely, the current study is confirmatory, adding to the list of lysosomal marker proteins reported in this context.

Below are specific comments.

1. NOVELTY: Most importantly, the novelty of this manuscript is significantly compromised by previous publications. The relationship between increased lysosomal biogenesis and microglial activation has been investigated in TBI models and very similar results have been obtained (Tanaka et al, 2013, Neuroscience). PGRN is well known to be increased near Abeta plaques in AD mouse models and human AD (Baker et al, 2006, Nature; Pereson et al, 2009, J Pathol; Minami et al, 2014, Nat Med; Gowrishankar et al, 2015, PNAS; Mendsaikhan et al, 2019, Acta Neuropathol Commun). Reduced Abeta burden and increased microglial response near Abeta plaques have been already reported in previous studies (Takahashi et al, 2017, Acta Neuropathol; Hosokawa et al, 2018 Exp Anim; Götzl et al, 2019, EMBO Mol Med). A previous study has shown partial DAM signature in Grn^{-/-} microglia (Götzl et al, 2019, EMBO Mol Med).

Reply: We agree with the reviewers that there have been several published studies on the role of PGRN in AD already. However, conflicting results have been reported regarding the effect of PGRN on A-beta plaque dynamics in AD mouse models (Minami et al., 2014; Van Kampen and Kay, 2017; Hosokawa et al., 2018; Takahashi et al., 2017). Our new results showing that age and sex have a significant effect on the role of PGRN in APP levels will shed light on some of discrepancies observed in these earlier studies.

Additionally, our results demonstrated that PGRN deficiency and the presence of A-beta have synergistic effect on microglial activation. We've examined the expression pattern of several DAM markers, including Galectin-3, GPNMB and TREM2 and have found that they are specifically activated in the microglia near A-beta plaques in PGRN deficient mice. Their expression pattern overlaps with several lysosomal proteins examined, suggesting that lysosomal abnormalities are correlated with microglial activation in response to PGRN loss.

2. CONTEXT: In the Introduction, the background of PGRN is described in detail, but its role Alzheimer's disease is completely lacking. Numerous previous studies investigating the role of PGRN in AD are not reviewed. It is essential that the Introduction and to some extent the Abstract make clear the extensive existing information, and indicate what new question is solved by repeating the previous work with a different AD strain. The literature cannot be ignored until the reader reaches the Discussion.

Reply: Thanks for your suggestion. We added related references, re-organized and modified these parts accordingly.

3. FUNCTION: The paper documents in detail altered protein level and distribution for many lysosomal constituents in the Grn deficient 5XFAD condition. However, there is

essentially no study of function. For the mice, Abeta plaque is decreased in males and unchanged in females. No change in behavior assessment of the 5XFAD mice is detected. At the cellular level, no assays of cell function such as phagocytosis or lysosomal degradation for Abeta or other molecules are assayed. The few pieces of information in the manuscript suggest no major change in mouse or cell function. The focus is specifically on changes in the expression of numerous lysosomal markers. Any statement that this manuscript documents lysosomal "dysfunction" needs to be deleted from Abstract, Introduction, Results and Discussion.

Reply: Thanks for your suggestion. We have revised our manuscript accordingly.

4. SEX: The finding of a male-specific change in Abeta plaques by PGRN deficiency is intriguing, but this observation is simply reported, and then not studied at all. Whether Abeta plaques are changed by PGRN loss at older ages (e.g., 10 m) in male and female remains unclear. Did the authors use primary microglia from female versus male mice? What is molecular basis for this difference?

Reply: Thank you for your suggestion. In the re-submission, we have performed ELISA to measure A-beta levels in both soluble and insoluble fractions in 4.5 month (Fig. 2A) and 10 month (Fig. 3B) old male and female mice. In addition, we have analyzed A-beta pathology in both male and female mice at 10-month-old (Fig. 3A). The results from the ELISA are consistent with our quantification of A-beta immuno-staining intensity and suggest that the production of A-beta is reduced in the young male mice upon PGRN loss. To determine the molecular mechanism underlying this sex specific difference, we measured the levels of APP precursor protein using immunoblot and showed a significant reduction in the levels of human mutant APP in the 4.5-month-old male *Grn*^{-/-} mice (Fig. 2B, 2C). No significant changes in the levels of presenilin1 (PS1), BACE1 or mouse APP, were observed in these mice (Fig. 2B-C, Fig. S3), suggesting that PGRN might regulate the metabolism of mutant human APP in a sex specific manner.

In addition, we have determined microglial responses to A β plaques in both female and male (Fig. 4F and 9B) mice using Galectin-3 and cathepsin D as markers and did not observe any sex specific difference in microglial responses to A β plaques. Thus, for the in vitro assays, we have used mixed male and female microglia.

5. In DAM state, microglia express hundreds of DAM signature genes (Keren-Shaul et al, 2017, Cell). Only a few were shown to be increased in the present study and more typical DAM genes are not shown to be upregulated. It is thus unclear whether PGRN deficiency induces DAM state or another undefined state in microglia near Ab plaques. The authors should tone down and avoid stating that PGRN loss results in DAM gene expression.

Reply: Thanks for your suggestion. We have modified the text accordingly.

6. In transcriptomic analysis of the present study, DEGs in 4.5m GRN^{-/-} and 5XFAD vs WT have not been detected, which seems inconsistent with previous studies and the fact that there exist a plenty of amyloid plaques in 5XFAD at this age. It might have something to do with FC threshold of the analysis. But because of this discrepancy, transcriptomic changes in 4 genotypes are not convincing and requires re-analysis and/or detailed explanation.

Reply: We agree with the reviewer's concerns. This might be due to variabilities between samples and the low number of samples we have used in the RNA seq analysis.

However, the differences we have seen between 5XFAD and 5XFAD Grn^{-/-} samples are obvious and significant in our analysis.

7. In Figure 8D and 8E, the result of TFE3 translocation is less than convincing. It looks as though the PGRN deficiency causes just an increase, but not nuclear translocation, of TFE3 under the current magnification. The authors should use higher magnification for this analysis and nuclear/cytoplasmic TFE3 ratio should be calculated as shown in in vitro studies in Figure 11D. In addition, the correlation with an increase in galectin-3 should be quantitatively assessed.

Reply: We agree with the reviewer that it will be nice to show nuclear/cytoplasmic ratio of TFE3 for tissue section staining as well. However, technically it's very challenging to determine the intensity of TFE3 in the cytosol due to the small size of microglia. Thus, we quantified nuclear TFE3 intensity to determine whether this pathway is activated in the 5XFAD Grn^{-/-} mice. We also agree that it will be nice to show a correlation between the increase in TFE3 nuclear signals and the increase in galectin-3 signals. In our studies, we've found that not all the microglia with high TFE3 nuclear signals show upregulation of galectin-3. However, it should be noted that there are other homologs of TFE3, such as TFEB and MiTF, which we cannot measure their levels in microglia due to the availabilities of specific antibodies that can be used in tissue section staining.

8. It is stated that "PGRN deficiency in microglia results in impairment of lysosomal membrane integrity and increased inflammatory responses after phagocytosis of Abeta fibrils". However, the statement is not supported by the authors' results. First, phagocytosis of fibrils has not been assessed in this study. It is unlikely that the fibrils without sonication are properly phagocytosed. Second, there are no data showing Abeta induces impairment of lysosomal membrane integrity. It looks like Abeta just increases galectin-3 and the colocalization analysis shown in Figure 9 has not been performed.

Reply: Thanks for your careful review of our manuscript. We totally agree and have revised the description of our results accordingly.

9. In Figure 9, quantification of untreated groups is missing. It is thus unclear whether LLOME is effective in Ctrl/WT.

Reply: Thanks for the suggestion. Since there is very little overlap between galectin-3 and LAMP1 signals in untreated group and no obvious differences between WT and Grn^{-/-} cells, we focused our quantification on the LLOME treated samples.

10. In all bar graphs, data points should be presented using dot plots to show the distribution. In addition, the authors should describe how normality was estimated in the method.

Reply: Thanks for the suggestions. We have changed all the graphs to dot plots and have added detailed description on quantification and normalization methods in the figure legend.

11. In Figure 3B, 10 and 11, a t-test is inappropriately used for the statistical analysis. the authors should use one-way or two-way ANOVA for the results.

Reply: Thanks for the suggestions. We have reanalyzed the data using one-way ANOVA or two-way ANOVA.

Minor comments:

12. It would be helpful if the various panels and graphs were labeled with age and sex of animals and the brain region analyzed.

Reply: Thanks for the suggestions. We have added the labels in the figures.

13. In Figure 8D and 11C, while Dapi appeared to be used for nuclei staining, the method says Hoechst was used.

Reply: Thanks for the suggestion. We modified the figures accordingly.

14. In Figure 11D, it looks like the data are normalized to Grn^{-/-} treated with Abeta but it is not described anywhere.

Reply: Sorry for the confusion. In Figure 11D, Y axis show the ratio of nuclear to cytoplasmic TFE3 intensity. It is not normalized to any group.

Reviewer #2 (Comments to the Authors (Required)):

In this manuscript, the authors report that PGRN deficient decreases Abeta plaques in male, not female, 5XFAD mice. Further, the authors claim that PGRN regulates microglia-mediated inflammation through modulating lysosomal function in response to Amyloid beta plaques. The paper is well written and easy to understand. However, some critical primary data is not carefully performed making the overall interpretation of the data uncertain. These major concerns need to be addressed to make this paper suitable for publication.

1) The most major concern are the experiments quantifying Abeta plaque pathology in Fig. 2. In general, they are poorly done and Immunofluorescence is not the best approach for quantifying Amyloid plaque burden. Also, 5X FAD mice have a substantial amount of variability in plaque number and size at 4.5 months of age. Much larger numbers of mice per group are necessary to draw any solid conclusion. General experimental details are missing. For example, how many sections per mice were measured? How far apart were the slices?

Reply: Thanks for the suggestion. Yes, we totally agree with you that 5X FAD mice have a substantial amount of variability in plaque number and size. All the Immunofluorescence staining, three brain sections per mouse, separated by 100 μ m were used for quantification. The mean from the three sections was used as the value for each mouse. The detailed method can be found in the Material and Methods section. In addition, we have performed ELISA analysis to determine the levels of soluble and insoluble A-beta and obtained consistent results as the abeta plaques quantification in both 4.5 month (Fig 2A) and 10 month (Fig 3B) old male and female mice.

2) Further, the 6E10 staining simply does not look right. IHC with antigen retrieval (Formic Acid or PAF) needs to be performed. Finally, a complementary approach is necessary to draw definitive conclusions. Specifically, the inclusion of ELISA measurements of soluble/insoluble Abeta species would allow a precise quantitative measurement.

Reply: Thanks for the suggestion. Yes, we have added Abeta quantification by ELISA (Fig.2A and Fig 3B) which includes both soluble and insoluble Abeta40 and Abeta42.

3) Considering that most of the conclusions are based on immunofluorescence assays, it is essential to include more negative controls for them (i.e. secondary antibodies alone to see how much of the signal could be background). For instance, Fig 2, panel A shows the staining of 6E10 in brain tissue, and the background is very high and the morphology of the "plaque" doesn't look normal. Another example is in Fig 1, panel A, B, and C with PGRN and 6E10 antibodies. This control is important because one of the main conclusions of this work is that PGRN and microglia are recruited to Abeta plaques. Other antibodies with questionable staining are TREM2 (Fig5-G), galectin 3 (fig4-E), CathB (Fig7-C).

Reply: Thanks for the comments. We have performed both ThioS and 6E10 staining on the same sections and shown that 6E10 can recognize the A-beta plaque specifically. We have been using the anti-PGRN antibodies for over 10 years and have confirmed the specificity using the KO samples (Zhou et al, JCB 2016). We've tested the galectin-3 and TREM2 antibodies in Western blot analysis and have observed specific signals at bands of correct molecular weight. In addition, the band intensity in Western blot analysis is correlated with changes in the staining intensity. Thus we are pretty confident about the specificities of these two antibodies, although unfortunately we do not have KO samples to confirm. All the cathepsin antibodies have been verified using KO samples. We have added these verification data as Fig. S7.

4) On page 4 the authors claim that Abeta plaque accumulation decreases the efficiency of PGRN processing based only on a western blot in figure 1 panel D, where they compare levels of full PGRN and GRNs in WT and 5xFAD GRN^{-/-}. It could be interesting to see if treatment with Abeta can impair PGRN processing to make this claim stronger.

Reply: Thanks for your suggestion. We completely agree with you. We have removed PGRN processing from this manuscript since we are going to characterize the changes in more detail using antibodies specific to each granulin peptide, which will be part of another manuscript.

5) What antibody was used to detect GRNs? Does the antibody recognize all the different GRN peptides?

Reply: We used sheep anti-PGRN (R&D Systems, AF2557) to detect both PGRN and GRNs. Based on our overexpression assay, this antibody preferentially recognizes GrnB, GrnC and GrnF. However, we've decided to perform more detailed analysis on the granulin peptides using antibodies specific to each granulin peptide we've recently generated to be included in another study.

6) In figure 3 panel C, it is not clear what IBA1 levels near Ab plaques mean. Just the fluorescence intensity of cells near Ab plaques? How was this data normalized? This question is because it is possible to see an increase of IBA 1 signal in all the picture, not only near Ab plaques, so the data that is has been reported with this graph is not necessarily indicating the recruitment of IBA1 (microglia) to the plaques as a specific

response but just an increased pool of activated microglia. So maybe, results should be normalized by the total number of IBA1+ microglia. Same for CD68, CatD, and CatB
Reply: Thanks for your suggestion. Our data showing enhanced microglial levels around abeta plaques in the absence of PGRN is consistent with a previous study (Götzl et al, 2019, EMBO Mol Med).

For the quantitative analysis of IBA1 levels near A β plaques (new Fig. 6E), the A β plaques were first selected using the ROI tool in Image J. Then all IBA1-positive microglia that fully or partially overlap with the A β plaques signal were selected (Analyze\tools\ROI manager), Subsequently, the fluorescence area and intensity were measured. For CD68, Galectin-3, GPNMB, TREM2, cathepsin B (CathB) and cathepsin D (CathD) levels near A β plaques, the images were analyzed using the same procedure in ImageJ. All these data normalized to 5XFAD group.

We have also quantified the total IBA1 and CD68 levels (new Fig. 6D) and we agree with the reviewer that there is overall microglia activation and increased microglial numbers in PGRN deficient mice and thus we cannot claim that there is increased recruitment of microglia to the plaque. We've modified the text accordingly. However, all the markers associated with microglial activation, including Galectin-3, and GPNMB are expressed at much higher levels in the microglia near the plaques. The CD68 signals are also more enriched in the microglia near the plaques although there is a global activation of microglia (comparing Fig. 6D and 6E).

7) In Fig. 5 E, a higher magnification should be included for GPNMB staining to enable visualization. Quantification of co-localization could also be performed. Also, in panel G, a lower magnification for TREM2 to have a broad idea of the expression of this protein in the sample.

Reply: Thanks for your suggestion. We added high magnification image of GPNMB (new Fig. 5E), and low magnification image of TREM2 (new Fig. 6A).

8) On page 6, the authors claim that the data support the idea that "PGRN is critical for proper lysosome function in microglia in response to Abeta plaques." However, data only shows an increase in the levels of lysosomal proteins. What about actual function? How does Abeta affect the activity of lysosomal enzymes, such as CatD and B. This data could help explain how Abeta affects PGRN and lysosome function.

Reply: Thanks for your suggestion. We have revised our manuscript to describe our results more accurately.

9) In Fig. 8 panel D. It would be helpful to add a merged image of TFE3/DAPI with IBA1 and galectin-3 to visualize the data in a better way. How do authors know which cells are close to Abeta plaques in this figure? How is nuclear translocation normalized?

Reply: Thanks for your suggestion. However, since we have both Hoescht and Galectin-3 in blue colors, it will be very hard to distinguish them in the merged image, so we decided not do show the merged image. We located abeta plaques by the clustered IBA1 signals since we don't have additional channel for A-beta staining. TFE3 nuclear intensity was normalized to 5XFAD group (set nuclear TFE3 intensity of 5XFAD group as 1).

10) Figure 10 and 11. Considering that there is an increase in the levels of GPNMB,

which can be seen in the figure but not in the graph, and CatD in untreated GRN^{-/-} cells, and also an increased translocation of TFE3 to the nucleus, it would be useful to graph only the ratio between WT/GRN^{-/-} cells untreated and treated with Abeta. This comparison would allow readers to see if these changes are produced by the deficiency of PGRN or from the addition of Abeta.

Reply: Thank you for your suggestion. We have included more representative images in Figs 10 and 11 (new Figs 11 and 12), and re-analyzed these data by two-way ANOVA. We have also analyzed the ratio of cathepsin D, Galectin-3 and GPNMB signals in WT and Grn^{-/-} cells in untreated and a-beta treated conditions as suggested (new Figs 11 and 12).

11) The co-localization of PGRN and Abeta has been previously studied and associated with lysosomal proteins and microglia. For instance, Mendsaikhan et al., 2019 10.1177/1533317509346209. It would be interesting to discuss more in-depth how this new work contributes to the field compared with the existing published data.

Reply: Thank you for your suggestion. We have cited this paper in our manuscript and edited the introduction and discussion part accordingly.

12) Moreover, other papers indicate that deficiency of PGRN has a protective role against Abeta accumulation (Hosokawa et al., 2018. 10.1538/expanim.17-0060; Takahashi et al., 2017 10.1007/s00401-017-1668-z). Do you think this is also associated with gender? How do these previous publications affect the conclusion of this work?

Reply: Thank you for your suggestion. Hosokawa et al., 2018, they used three female APP mice and three female APP/Grn^{+/-} mice (age range: 16–18-month-old) in this study. Abeta plaques analysis showed that the number and area of A β plaque was significantly decreased in APP/Grn^{+/-} mice as compared to APP mice. Their data suggest that PGRN haploinsufficiency may decrease accumulation of A β .

Takahashi et al., 2017, 6-month-old male and 16-month-old APP/PS1, APP/PS1 Grn^{+/-}, and APP/PS1 Grn^{-/-} mice were used (n=3-5). Anti-A β antibody staining shows no significant differences between Grn genotypes in 6-month-old mice, but reveals a significant reduction in the area of diffuse A β plaque pathology in 16-month-old APP/PS1 Grn^{-/-} mice.

In our case, we used a different mouse model-5XFAD mouse model, which is a rapid-onset amyloid plaque models and develops plaques at 2 months. Our data showed a significant reduction of A β in the 4.5-month-old male 5XFAD Grn^{-/-} mice, but not in the female 5XFAD Grn^{-/-} mice, resulting from a reduction in the APP levels. In the 10-month-old, PGRN loss results in a mild decrease of plaque area and numbers without affecting total intensity. Our results from the 10-month-old are somewhat consistent with previous published reports, showing that PGRN deficiency can decrease the number and area of A β plaque (Takahashi et al., 2017; Hosokawa et al., 2018). However, Takahashi et al., 2017 did not find any alteration in APP or A-beta levels the 6-month old male mice, which could be due to the difference in the APP variant expressed, the age of mice used and the amount of APP produced.

In addition, a human genetic study examining the association between granulin gene polymorphisms and Alzheimer's disease in the Finnish population has shown that GRN

polymorphisms may influence AD risk in a male-specific manner (Viswanathan, J. *et al.* 2009), which might be explained by our findings. We have added some of these points in our discussion.

13) Finally how do you account for the discrepancy between your work and Minami et al, Nature Medicine, which convincingly shows Progranulin protects against amyloid β deposition and toxicity in Alzheimer's disease mouse models?

Reply: In Minami et al, 2014 paper, the selective ablation of 50% PGRN in microglia in J20 (APP^{high}) with abundant amyloid loading was shown to impair phagocytosis, increase plaque load and exacerbate cognitive deficits, while global ablation of PGRN did not affect amyloid levels in J20^{low} model with sparse plaque deposition. Lentivirus-mediated PGRN overexpression lowered plaque load in aged 5xFAD mice. There are several reasons that might explain the different results we've obtained. First, mouse model is different. To test the effect of PGRN loss, they used J20 mice expressing APP^{swe, ind} mutation. We used 5XFAD mouse model (5XFAD mice express human APP and Presenilin1 transgenes with a total of five AD-linked mutations) which is a rapid-onset amyloid plaque models and develops plaques at 2 months. Second, different PGRN-deficient mouse lines are used. They have used whole body PGRN knockout in the APP^{low} mice and didn't find any effect of PGRN on APP and A-beta. Microglia specific reduction of PGRN using the LysM-Cre system shows exacerbated A-beta pathology. In our case, we generated whole body PGRN knockout mice (Yin et al., 2010). The reduction in the human mutant APP levels is likely due to loss of PGRN in neurons, rather than microglia. Third, the age of mice examined might also explain some of the phenotypic differences, since the A-beta pathology is very dynamic and different results can be obtained at different stages. In their study with the APP^{low} mice, 9-13 months old mixed male and females were used to examine the effect of PGRN on APP and A-beta. The effect of microglia on A-beta plaque and neuronal health can also be complicated by aging. We have added some of these points in the discussion.

Reviewer #3 (Comments to the Authors (Required)):

This study by Du et al investigates the role of progranulin in pathogenesis of Alzheimer's Disease. Loss-of-function mutations in the progranulin gene, most of which cause haploinsufficiency, are a major cause of frontotemporal dementia, but a polymorphism with milder effects on progranulin levels increases risk for AD. Prior work on progranulin in AD mouse models has yielded mixed results when AD mouse models are crossed with Grn KO mice.

This study is a useful addition to this literature and adds to the data showing possibly counterintuitive protective effects of progranulin deficiency using the 5X FAD model. The discovery of sex-specific effects of progranulin deficiency, and the careful study of microglia are strengths of this paper, particularly insofar as the data begin to connect progranulin's known lysosomal functions with the changes observed in GrnKO microglia exposed to amyloid beta. However, additional data tying the changes in microglia to amyloid pathology would strengthen this paper. Additional study of sex and age

differences in amyloid plaque load and microglial activation in the GrnKO:5XFAD cross might help clarify the association between microglial changes, amyloid plaques, and behavioral deficits.

Reply: We truly appreciate the encouraging comments.

1.) The sex difference in the effects of Grn KO is very interesting given the known sex differences in AD risk. It therefore seems that focusing solely on female mice beginning in figure 3 leaves potentially valuable data unexplored. Measuring microglial activation and localization around plaques in male and female mice of different ages might reveal mechanisms underlying the sex difference in GrnKO:5XFAD plaque burden. For example, do male and female GrnKO:5XFAD mice show differences in plaque load and microglial activation at 2 or 3 months? Perhaps the male mice would have greater microglial activation at these early ages that could result in greater clearance of amyloid?

Reply: Thanks for your helpful suggestions. The sex difference of PGRN is very intriguing. We added ELISA analysis to confirm abeta plaques quantification in both 4.5 month (Fig. 2A) and 10 month (Fig 3B) old mice. Immunoblot analysis revealed a significant reduction in the levels of APP in the 4.5-month-old male mice, but no significant changes in the levels of presenilin1 (PS1) and BACE1 in these mice (Fig 2B-C), suggesting that the reduction of A β plaques deposition in the 4.5-month-old male mice is due to changes of the metabolism of human mutant APP caused by PGRN loss.

We have analyzed microglial phenotypes around A-beta plaques in both male and female mice and did not see any obvious differences (Fig. 4F, Galectin-3 staining; Fig. 9B, cathepsin D staining). In addition, in the 10-month-old mice, PGRN loss does not cause a main difference in the A-beta pathology between male and female mice. We conclude that the sex difference is due to a specific effect of PGRN on mutant human APP in the young male mice.

2.) Were any sex differences in microglia observed in GrnKO mice that didn't carry the 5XFAD transgene?

Reply: So far we have not seen any sex differences of PGRN in microglia without 5XFAD transgene. We think the sex difference is largely caused by a male specific effect of PGRN on the mutant APP expressed from the transgene in neurons.

3.) Female GrnKO:5XFAD mice show striking changes in microglia, but no change in amyloid plaque load or behavior versus Grn WT:5XFAD mice. While there is extensive data showing the importance of microglia in AD pathogenesis, what is the data from this study to establish a link between microglial changes due to progranulin KO and amyloid pathology or behavioral deficits?

Reply: Thank you. These are excellent questions! The role of microglia in A-beta pathology is very complicated and still not well understood. Microglia can internalize A β but not necessarily degrade A β due to poor lysosomal acidification. Some studies have shown that microglial activities are important for the development of A-beta plaques, while other ones have shown that microglial activation regulates the conversion between diffuse vs dense core plaques. Although PGRN loss do not have a major effect on A β dynamics,

it does seem to reduce the number and area of A β plaques without affecting overall A β levels at 10 month-old, indicating that A β is more concentrated at each plaque.

In response to A β , microglia also undergo activation, which can lead to the secretion of toxic cytokines and aberrant pruning of synapses. PGRN deficiency has been shown to result in increased microglial activation and synaptic pruning, which could have detrimental effects in AD mouse models. Since PGRN deficient male mice have reduced A-beta and enhanced microglial activation at the same time, it is very hard to predict outcomes in behavioral tests. Our results do show a trend of improved behavior in the Y-maze in young PGRN knockout males compared to age matched controls, but it didn't reach statistical significance.

MS: LSA-2020-00874-T

Dr. Fenghua Hu
Cornell University
Weill Institute for Cell and Molecular Biology and Department of Molecular Biology and Genetics
345 Weill Hall, Cornell University
Ithaca, NY 14853

Dear Dr. Hu,

Thank you for submitting an appeal for your manuscript "Lysosome abnormalities caused by progranulin deficiency lead to enhanced inflammatory responses to amyloid-beta plaques" that was previously reviewed at Life Science Alliance (LSA).

We have now evaluated your appeal letter, revised manuscript and the point-by-point rebuttal and I am pleased to let you know that we deem the manuscript sufficiently revised to send back to the referees. Of course, the final decision on whether the revisions satisfactorily address the reviewers' concerns will have to be determined by them.

We ask you to submit the revised manuscript back to us using the link below so that we can send it back to the referees for re-review.

<https://lsa.msubmit.net/cgi-bin/main.plex?el=A7Na2Wi4A7JeG6l4B9ftdZIsPwW4xNolfoW2gWqj8gZ>

Yours sincerely,

Shachi Bhatt, Ph.D.

Executive Editor

Life Science Alliance

<https://www.lsjournal.org/>

Interested in an editorial career? EMBO Solutions is hiring a Scientific Editor to join the international Life Science Alliance team. Find out more here -

https://www.embo.org/documents/jobs/Vacancy_Notice_Scientific_editor_LSA.pdf

We thank the editor and reviewers for their constructive comments regarding our manuscript. During the past 5 months, we have collected new data and revised our manuscript extensively to address the reviewers' and the editor's concerns.

Editor's comments:

Given the interest in the topic, I would be open to resubmission to Life Science Alliance of a significantly revised and extended manuscript that addresses most of the reviewers' concerns and is subject to further peer-review, preferably by the same set of referees. The concern about conceptual advance (Rev 1) and behavioral analysis (Rev 1 and 2) and figuring out the mechanism of sex differences (Rev 1 and 3) do not need to be addressed for a publication in Life Science Alliance. All other concerns, particularly the ones pertaining to the effects of progranulin deficiency on the cellular function of microglia and quantification of Abeta pathology, and others should be addressed for the manuscript to be reconsidered at Life Science Alliance.

Reply: We thank the editor for the constructive feedbacks. We have added the following new data in the revised manuscript to address the reviewers' concerns.

1. Quantification of A-beta levels by ELISA in the 4-month-old and 10-month-old male and female cortical lysates (new Fig. 2A and Fig. 3B) to show age and sex dependent regulation of A-beta by PGRN.
2. Quantification of A-beta pathology in the 10-month-old male and female PGRN deficient mice to show an age dependent regulation of A-beta deposition by PGRN (new Fig. 3A).
3. Western blot analysis to show a specific decrease in the levels of mutant human APP in the young male mice in the absence of PGRN (new Fig. 2B, Fig. S3).
4. Analysis of microglial responses near A-beta plaques in the male PGRN deficient mice to show that PGRN deficiency does not result in sex specific microglial responses to A-beta (new Fig. 4F and 9B).
5. New western blot using cathepsin KO samples to show the specificities of the antibodies used in the study (new Fig. S7).

Our new data demonstrates a novel role of PGRN in regulating APP metabolism in a sex and age dependent manner. In addition to these new exciting data, we have revised the text extensively to address each reviewer's comments. Please see detailed responses below.

Reviewer #1 (Comments to the Authors (Required)):

In the present study, Du et al. investigate the role of PGRN in microglial responses to Abeta plaques using 5XFAD mice and primary microglial culture. The results show that PGRN is upregulated in microglia near Abeta plaques. They also report that PGRN deficiency leads to a decrease in Abeta plaques in male 5XFAD mice at 4.5 months of age, enhanced microgliosis, and an increase in expression of some lysosomal and inflammatory proteins such as cathepsin B and TREM2 in microglia near Abeta plaques. In vitro studies using microglia culture support some results from their mouse studies.

Unfortunately, nearly all of the data obtained from the present study are already reported in very similar studies, or can be easily predicted from the previous publications. Numerous studies have reported on Grn^{-/-} crossed with transgenic AD mice, and it is not clear that the current study substantially advances the field. Largely, the current study is confirmatory, adding to the list of

lysosomal marker proteins reported in this context.

Below are specific comments.

1. NOVELTY: Most importantly, the novelty of this manuscript is significantly compromised by previous publications. The relationship between increased lysosomal biogenesis and microglial activation has been investigated in TBI models and very similar results have been obtained (Tanaka et al, 2013, Neuroscience). PGRN is well known to be increased near Abeta plaques in AD mouse models and human AD (Baker et al, 2006, Nature; Pereson et al, 2009, J Pathol; Minami et al, 2014, Nat Med; Gowrishankar et al, 2015, PNAS; Mendsaikhan et al, 2019, Acta Neuropathol Commun). Reduced Abeta burden and increased microglial response near Abeta plaques have been already reported in previous studies (Takahashi et al, 2017, Acta Neuropathol; Hosokawa et al, 2018 Exp Anim; Götzl et al, 2019, EMBO Mol Med). A previous study has shown partial DAM signature in Grn-/- microglia (Götzl et al, 2019, EMBO Mol Med).

Reply: We agree with the reviewers that there have been several published studies on the role of PGRN in AD already. However, conflicting results have been reported regarding the effect of PGRN on A-beta plaque dynamics in AD mouse models (Minami et al., 2014; Van Kampen and Kay, 2017; Hosokawa et al., 2018; Takahashi et al., 2017). Our new results showing that age and sex have a significant effect on the role of PGRN in APP levels will shed light on some of discrepancies observed in these earlier studies.

Additionally, our results demonstrated that PGRN deficiency and the presence of A-beta have synergistic effect on microglial activation. We've examined the expression pattern of several DAM markers, including Galectin-3, GPNMB and TREM2 and have found that they are specifically activated in the microglia near A-beta plaques in PGRN deficient mice. Their expression pattern overlaps with several lysosomal proteins examined, suggesting that lysosomal abnormalities are correlated with microglial activation in response to PGRN loss.

2. CONTEXT: In the Introduction, the background of PGRN is described in detail, but its role Alzheimer's disease is completely lacking. Numerous previous studies investigating the role of PGRN in AD are not reviewed. It is essential that the Introduction and to some extent the Abstract make clear the extensive existing information, and indicate what new question is solved by repeating the previous work with a different AD strain. The literature cannot be ignored until the reader reaches the Discussion.

Reply: Thanks for your suggestion. We added related references, re-organized and modified these parts accordingly.

3. FUNCTION: The paper documents in detail altered protein level and distribution for many lysosomal constituents in the Grn deficient 5XFAD condition. However, there is essentially no study of function. For the mice, Abeta plaque is decreased in males and unchanged in females. No change in behavior assessment of the 5XFAD mice is detected. At the cellular level, no assays of cell function such as phagocytosis or lysosomal degradation for Abeta or other molecules are assayed. The few pieces of information in the manuscript suggest no major change in mouse or cell function. The focus is specifically on changes in the expression of numerous lysosomal markers. Any statement that this manuscript documents lysosomal "dysfunction" needs to be deleted from Abstract, Introduction, Results and Discussion.

Reply: Thanks for your suggestion. We have revised our manuscript accordingly.

4. SEX: The finding of a male-specific change in Abeta plaques by PGRN deficiency is intriguing, but this observation is simply reported, and then not studied at all. Whether Abeta plaques are changed by PGRN loss at older ages (e.g., 10 m) in male and female remains unclear. Did the authors use primary microglia from female versus male mice? What is molecular basis for this difference?

Reply: Thank you for your suggestion. In the re-submission, we have performed ELISA to measure A-beta levels in both soluble and insoluble fractions in 4.5 month (Fig. 2A) and 10 month (Fig. 3B) old male and female mice. In addition, we have analyzed A-beta pathology in both male and female mice at 10-month-old (Fig. 3A). The results from the ELISA are consistent with our quantification of A-beta immuno-staining intensity and suggest that the production of A-beta is reduced in the young male mice upon PGRN loss. To determine the molecular mechanism underlying this sex specific difference, we measured the levels of APP precursor protein using immunoblot and showed a significant reduction in the levels of human mutant APP in the 4.5-month-old male *Grn*^{-/-} mice (Fig. 2B, 2C). No significant changes in the levels of presenilin1 (PS1), BACE1 or mouse APP, were observed in these mice (Fig. 2B-C, Fig. S3), suggesting that PGRN might regulate the metabolism of mutant human APP in a sex specific manner.

In addition, we have determined microglial responses to A β plaques in both female and male (Fig. 4F and 9B) mice using Galectin-3 and cathepsin D as markers and did not observe any sex specific difference in microglial responses to A β plaques. Thus, for the in vitro assays, we have used mixed male and female microglia.

5. In DAM state, microglia express hundreds of DAM signature genes (Keren-Shaul et al, 2017, Cell). Only a few were shown to be increased in the present study and more typical DAM genes are not shown to be upregulated. It is thus unclear whether PGRN deficiency induces DAM state or another undefined state in microglia near Ab plaques. The authors should tone down and avoid stating that PGRN loss results in DAM gene expression.

Reply: Thanks for your suggestion. We have modified the text accordingly.

6. In transcriptomic analysis of the present study, DEGs in 4.5m GRN^{-/-} and 5XFAD vs WT have not been detected, which seems inconsistent with previous studies and the fact that there exist a plenty of amyloid plaques in 5XFAD at this age. It might have something to do with FC threshold of the analysis. But because of this discrepancy, transcriptomic changes in 4 genotypes are not convincing and requires re-analysis and/or detailed explanation.

Reply: We agree with the reviewer's concerns. This might be due to variabilities between samples and the low number of samples we have used in the RNA seq analysis. However, the differences we have seen between 5XFAD and 5XFAD *Grn*^{-/-} samples are obvious and significant in our analysis.

7. In Figure 8D and 8E, the result of TFE3 translocation is less than convincing. It looks as though the PGRN deficiency causes just an increase, but not nuclear translocation, of TFE3 under the current magnification. The authors should use higher magnification for this analysis and nuclear/cytoplasmic TFE3 ratio should be calculated as shown in in vitro studies in Figure 11D. In addition, the correlation with an increase in galectin-3 should be quantitatively assessed.

Reply: We agree with the reviewer that it will be nice to show nuclear/cytoplasmic ratio of TFE3 for tissue section staining as well. However, technically it's very challenging to determine the intensity of TFE3 in the cytosol due to the small size of microglia. Thus, we quantified nuclear TFE3 intensity to determine whether this pathway is activated in the 5XFAD *Grn*^{-/-} mice. We also agree that it will be nice to show a correlation between the increase in TFE3 nuclear signals and the increase in galectin-3 signals. In our studies, we've found that not all the microglia with high TFE3 nuclear signals show upregulation of galectin-3. However, it should be noted that there are other homologs of TFE3, such as TFEB and MiTF, which we cannot measure their levels in microglia due to the availabilities of specific antibodies that can be used in tissue section staining.

8. It is stated that "PGRN deficiency in microglia results in impairment of lysosomal membrane integrity and increased inflammatory responses after phagocytosis of A β fibrils". However, the statement is not supported by the authors' results. First, phagocytosis of fibrils has not been assessed in this study.

It is unlikely that the fibrils without sonication are properly phagocytosed. Second, there are no data showing Abeta induces impairment of lysosomal membrane integrity. It looks like Abeta just increases galectin-3 and the colocalization analysis shown in Figure 9 has not been performed.

Reply: Thanks for your careful review of our manuscript. We totally agree and have revised the description of our results accordingly.

9. In Figure 9, quantification of untreated groups is missing. It is thus unclear whether LLOME is effective in Ctrl/WT.

Reply: Thanks for the suggestion. Since there is very little overlap between galectin-3 and LAMP1 signals in untreated group and no obvious differences between WT and Grn^{-/-} cells, we focused our quantification on the LLOME treated samples.

10. In all bar graphs, data points should be presented using dot plots to show the distribution. In addition, the authors should describe how normality was estimated in the method.

Reply: Thanks for the suggestions. We have changed all the graphs to dot plots and have added detailed description on quantification and normalization methods in the figure legend.

11. In Figure 3B, 10 and 11, a t-test is inappropriately used for the statistical analysis. the authors should use one-way or two-way ANOVA for the results.

Reply: Thanks for the suggestions. We have reanalyzed the data using one-way ANOVA or two-way ANOVA.

Minor comments:

12. It would be helpful if the various panels and graphs were labeled with age and sex of animals and the brain region analyzed.

Reply: Thanks for the suggestions. We have added the labels in the figures.

13. In Figure 8D and 11C, while Dapi appeared to be used for nuclei staining, the method says Hoechst was used.

Reply: Thanks for the suggestion. We modified the figures accordingly.

14. In Figure 11D, it looks like the data are normalized to Grn^{-/-} treated with Abeta but it is not described anywhere.

Reply: Sorry for the confusion. In Figure 11D, Y axis show the ratio of nuclear to cytoplasmic TFE3 intensity. It is not normalized to any group.

Reviewer #2 (Comments to the Authors (Required)):

In this manuscript, the authors report that PGRN deficient decreases Abeta plaques in male, not female, 5XFAD mice. Further, the authors claim that PGRN regulates microglia-mediated inflammation through modulating lysosomal function in response to Amyloid beta plaques. The paper is well written and easy to understand. However, some critical primary data is not carefully performed making the overall interpretation of the data uncertain. These major concerns need to be addressed to make this paper suitable for publication.

1) The most major concern are the experiments quantifying Abeta plaque pathology in Fig. 2. In general, they are poorly done and Immunofluorescence is not the best approach for quantifying Amyloid plaque burden. Also, 5X FAD mice have a substantial amount of variability in plaque number and size at 4.5 months of age. Much larger numbers of mice per group are necessary to draw any solid conclusion. General experimental details are missing. For example, how many sections per mice were measured? How far apart were the slices?

Reply: Thanks for the suggestion. Yes, we totally agree with you that 5X FAD mice have a substantial amount of variability in plaque number and size. All the Immunofluorescence staining, three brain sections per mouse, separated by 100 μ m were used for quantification. The mean from the three sections was used as the value for each mouse. The detailed method can be found in the Material and Methods section. In addition, we have performed ELISA analysis to determine the levels of soluble and insoluble A-beta and obtained consistent results as the abeta plaques quantification in both 4.5 month (Fig 2A) and 10 month (Fig 3B) old male and female mice.

2) Further, the 6E10 staining simply does not look right. IHC with antigen retrieval (Formic Acid or PAF) needs to be performed. Finally, a complementary approach is necessary to draw definitive conclusions. Specifically, the inclusion of ELISA measurements of soluble/insoluble Abeta species would allow a precise quantitative measurement.

Reply: Thanks for the suggestion. Yes, we have added Abeta quantification by ELISA (Fig.2A and Fig 3B) which includes both soluble and insoluble Abeta40 and Abeta42.

3) Considering that most of the conclusions are based on immunofluorescence assays, it is essential to include more negative controls for them (i.e. secondary antibodies alone to see how much of the signal could be background). For instance, Fig 2, panel A shows the staining of 6E10 in brain tissue, and the background is very high and the morphology of the "plaque" doesn't look normal. Another example is in Fig 1, panel A, B, and C with PGRN and 6E10 antibodies. This control is important because one of the main conclusions of this work is that PGRN and microglia are recruited to Abeta plaques. Other antibodies with questionable staining are TREM2 (Fig5-G), galectin 3 (fig4-E), CathB (Fig7-C).

Reply: Thanks for the comments. We have performed both ThioS and 6E10 staining on the same sections and shown that 6E10 can recognize the A-beta plaque specifically. We have been using the anti-PGRN antibodies for over 10 years and have confirmed the specificity using the KO samples (Zhou et al, JCB 2016). We've tested the galectin-3 and TREM2 antibodies in Western blot analysis and have observed specific signals at bands of correct molecular weight. In addition, the band intensity in Western blot analysis is correlated with changes in the staining intensity. Thus we are pretty confident about the specificities of these two antibodies, although unfortunately we do not have KO samples to confirm. All the cathepsin antibodies have been verified using KO samples. We have added these verification data as Fig. S7.

4) On page 4 the authors claim that Abeta plaque accumulation decreases the efficiency of PGRN processing based only on a western blot in figure 1 panel D, where they compare levels of full PGRN and GRNs in WT and 5xFAD GRN^{-/-}. It could be interesting to see if treatment with Abeta can impair PGRN processing to make this claim stronger.

Reply: Thanks for your suggestion. We completely agree with you. We have removed PGRN processing from this manuscript since we are going to characterize the changes in more detail using antibodies specific to each granulin peptide, which will be part of another manuscript.

5) What antibody was used to detect GRNs? Does the antibody recognize all the different GRN peptides?

Reply: We used sheep anti-PGRN (R&D Systems, AF2557) to detect both PGRN and GRNs. Based on our overexpression assay, this antibody preferentially recognizes GrnB, GrnC and GrnF. However, we've decided to perform more detailed analysis on the granulin peptides using antibodies specific to each granulin peptide we've recently generated to be included in another study.

6) In figure 3 panel C, it is not clear what IBA1 levels near Ab plaques mean. Just the fluorescence intensity of cells near Ab plaques? How was this data normalized? This question is because it is possible to see an increase of IBA 1 signal in all the picture, not only near Ab plaques, so the data that is has been reported with this graph is not necessarily indicating the recruitment of IBA1 (microglia) to the plaques as a specific response but just an increased pool of activated microglia. So maybe, results should be normalized by the total number of IBA1+ microglia. Same for CD68, CatD, and CatB

Reply: Thanks for your suggestion. Our data showing enhanced microglial levels around abeta plaques in the absence of PGRN is consistent with a previous study (Götzl et al, 2019, EMBO Mol Med).

For the quantitative analysis of IBA1 levels near A β plaques (new Fig. 6E), the A β plaques were first selected using the ROI tool in Image J. Then all IBA1-positive microglia that fully or partially overlap with the A β plaques signal were selected (Analyze\tools\ROI manager), Subsequently, the fluorescence area and intensity were measured. For CD68, Galectin-3, GPNMB, TREM2, cathepsin B (CathB) and cathepsin D (CathD) levels near A β plaques, the images were analyzed using the same procedure in ImageJ. All these data normalized to 5XFAD group.

We have also quantified the total IBA1 and CD68 levels (new Fig. 6D) and we agree with the reviewer that there is overall microglia activation and increased microglial numbers in PGRN deficient mice and thus we cannot claim that there is increased recruitment of microglia to the plaque. We've modified the text accordingly. However, all the markers associated with microglial activation, including Galectin-3, and GPNMB are expressed at much higher levels in the microglia near the plaques. The CD68 signals are also more enriched in the microglia near the plaques although there is a global activation of microglia (comparing Fig. 6D and 6E).

7) In Fig. 5 E, a higher magnification should be included for GPNMB staining to enable visualization. Quantification of co-localization could also be performed. Also, in panel G, a lower magnification for TREM2 to have a broad idea of the expression of this protein in the sample.

Reply: Thanks for your suggestion. We added high magnification image of GPNMB (new Fig. 5E), and low magnification image of TREM2 (new Fig. 6A).

8) On page 6, the authors claim that the data support the idea that "PGRN is critical for proper lysosome function in microglia in response to Abeta plaques." However, data only shows an increase in the levels of lysosomal proteins. What about actual function? How does Abeta affect the activity of lysosomal enzymes, such as CatD and B. This data could help explain how Abeta affects PGRN and lysosome function.

Reply: Thanks for your suggestion. We have revised our manuscript to describe our results more accurately.

9) In Fig. 8 panel D. It would be helpful to add a merged image of TFE3/DAPI with IBA1 and galectin-3

to visualize the data in a better way. How do authors know which cells are close to Abeta plaques in this figure? How is nuclear translocation normalized?

Reply: Thanks for your suggestion. However, since we have both Hoescht and Galectin-3 in blue colors, it will be very hard to distinguish them in the merged image, so we decided not to show the merged image. We located Abeta plaques by the clustered IBA1 signals since we don't have an additional channel for A-beta staining. TFE3 nuclear intensity was normalized to the 5XFAD group (set nuclear TFE3 intensity of 5XFAD group as 1).

10) Figure 10 and 11. Considering that there is an increase in the levels of GPNMB, which can be seen in the figure but not in the graph, and CatD in untreated GRN^{-/-} cells, and also an increased translocation of TFE3 to the nucleus, it would be useful to graph only the ratio between WT/GRN^{-/-} cells untreated and treated with Abeta. This comparison would allow readers to see if these changes are produced by the deficiency of PGRN or from the addition of Abeta.

Reply: Thank you for your suggestion. We have included more representative images in Figs 10 and 11 (new Figs 11 and 12), and re-analyzed these data by two-way ANOVA. We have also analyzed the ratio of cathepsin D, Galectin-3 and GPNMB signals in WT and Grn^{-/-} cells in untreated and A-beta treated conditions as suggested (new Figs 11 and 12).

11) The co-localization of PGRN and Abeta has been previously studied and associated with lysosomal proteins and microglia. For instance, Mendsaikhan et al., 2019 10.1177/1533317509346209. It would be interesting to discuss more in-depth how this new work contributes to the field compared with the existing published data.

Reply: Thank you for your suggestion. We have cited this paper in our manuscript and edited the introduction and discussion part accordingly.

12) Moreover, other papers indicate that deficiency of PGRN has a protective role against Abeta accumulation (Hosokawa et al., 2018. 10.1538/expanim.17-0060; Takahashi et al., 2017 10.1007/s00401-017-1668-z). Do you think this is also associated with gender? How do these previous publications affect the conclusion of this work?

Reply: Thank you for your suggestion. Hosokawa et al., 2018, they used three female APP mice and three female APP/Grn^{+/-} mice (age range: 16–18-month-old) in this study. Abeta plaque analysis showed that the number and area of A β plaque was significantly decreased in APP/Grn^{+/-} mice as compared to APP mice. Their data suggest that PGRN haploinsufficiency may decrease accumulation of A β .

Takahashi et al., 2017, 6-month-old male and 16-month-old APP/PS1, APP/PS1 Grn^{+/-}, and APP/PS1 Grn^{-/-} mice were used (n=3-5). Anti-A β antibody staining shows no significant differences between Grn genotypes in 6-month-old mice, but reveals a significant reduction in the area of diffuse A β plaque pathology in 16-month-old APP/PS1 Grn^{-/-} mice.

In our case, we used a different mouse model-5XFAD mouse model, which is a rapid-onset amyloid plaque model and develops plaques at 2 months. Our data showed a significant reduction of A β in the 4.5-month-old male 5XFAD Grn^{-/-} mice, but not in the female 5XFAD Grn^{-/-} mice, resulting from a reduction in the APP levels. In the 10-month-old, PGRN loss results in a mild decrease of plaque area and numbers without affecting total intensity. Our results from the 10-month-old are somewhat consistent with previous published reports, showing that PGRN deficiency can decrease the number and area of A β plaque (Takahashi et al., 2017; Hosokawa et al., 2018). However, Takahashi et al., 2017 did not find any alteration in APP or A-beta levels in the 6-month-old male mice, which could be due to the difference in the APP variant expressed, the age of mice used and the amount of APP produced.

In addition, a human genetic study examining the association between granulin gene polymorphisms and Alzheimer's disease in the Finnish population has shown that *GRN* polymorphisms may influence AD risk in a male-specific manner (Viswanathan, J. *et al.* 2009), which might be explained by our findings.

We have added some of these points in our discussion.

13) Finally how do you account for the discrepancy between your work and Minami et al, Nature Medicine, which convincingly shows Progranulin protects against amyloid β deposition and toxicity in Alzheimer's disease mouse models?

Reply: In Minami et al, 2014 paper, the selective ablation of 50% PGRN in microglia in J20 (APP^{high}) with abundant amyloid loading was shown to impair phagocytosis, increase plaque load and exacerbate cognitive deficits, while global ablation of PGRN did not affect amyloid levels in $J20^{low}$ model with sparse plaque deposition. Lentivirus-mediated PGRN overexpression lowered plaque load in aged 5xFAD mice. There are several reasons that might explain the different results we've obtained. First, mouse model is different. To test the effect of PGRN loss, they used J20 mice expressing $APP^{swe, ind}$ mutation. We used 5XFAD mouse model (5XFAD mice express human APP and Presenilin1 transgenes with a total of five AD-linked mutations) which is a rapid-onset amyloid plaque model and develops plaques at 2 months. Second, different PGRN-deficient mouse lines are used. They have used whole body PGRN knockout in the APP^{low} mice and didn't find any effect of PGRN on APP and A-beta. Microglia specific reduction of PGRN using the LysM-Cre system shows exacerbated A-beta pathology. In our case, we generated whole body PGRN knockout mice (Yin et al., 2010). The reduction in the human mutant APP levels is likely due to loss of PGRN in neurons, rather than microglia. Third, the age of mice examined might also explain some of the phenotypic differences, since the A-beta pathology is very dynamic and different results can be obtained at different stages. In their study with the APP^{low} mice, 9-13 months old mixed male and females were used to examine the effect of PGRN on APP and A-beta. The effect of microglia on A-beta plaque and neuronal health can also be complicated by aging. We have added some of these points in the discussion.

Reviewer #3 (Comments to the Authors (Required)):

This study by Du et al investigates the role of progranulin in pathogenesis of Alzheimer's Disease. Loss-of-function mutations in the progranulin gene, most of which cause haploinsufficiency, are a major cause of frontotemporal dementia, but a polymorphism with milder effects on progranulin levels increases risk for AD. Prior work on progranulin in AD mouse models has yielded mixed results when AD mouse models are crossed with Grn KO mice.

This study is a useful addition to this literature and adds to the data showing possibly counterintuitive protective effects of progranulin deficiency using the 5X FAD model. The discovery of sex-specific effects of progranulin deficiency, and the careful study of microglia are strengths of this paper, particularly insofar as the data begin to connect progranulin's known lysosomal functions with the changes observed in GrnKO microglia exposed to amyloid beta. However, additional data tying the changes in microglia to amyloid pathology would strengthen this paper. Additional study of sex and age differences in amyloid plaque load and microglial activation in the GrnKO:5XFAD cross might help clarify the association between microglial changes, amyloid plaques, and behavioral deficits.

Reply: We truly appreciate the encouraging comments.

1.) The sex difference in the effects of Grn KO is very interesting given the known sex differences in AD risk. It therefore seems that focusing solely on female mice beginning in figure 3 leaves potentially valuable data unexplored. Measuring microglial activation and localization around plaques in male and female mice of different ages might reveal mechanisms underlying the sex difference in GrnKO:5XFAD plaque burden. For example, do male and female GrnKO:5XFAD mice show differences in plaque load and microglial activation at 2 or 3 months? Perhaps the male mice would have greater microglial activation at these early ages that could result in greater clearance of amyloid?

Reply: Thanks for your helpful suggestions. The sex difference of PGRN is very intriguing. We added ELISA analysis to confirm A β plaques quantification in both 4.5 month (Fig. 2A) and 10 month (Fig 3B) old mice. Immunoblot analysis revealed a significant reduction in the levels of APP in the 4.5-month-old male mice, but no significant changes in the levels of presenilin1 (PS1) and BACE1 in these mice (Fig 2B-C), suggesting that the reduction of A β plaques deposition in the 4.5-month-old male mice is due to changes of the metabolism of human mutant APP caused by PGRN loss.

We have analyzed microglial phenotypes around A-beta plaques in both male and female mice and did not see any obvious differences (Fig. 4F, Galectin-3 staining; Fig. 9B, cathepsin D staining). In addition, in the 10-month-old mice, PGRN loss does not cause a main difference in the A-beta pathology between male and female mice. We conclude that the sex difference is due to a specific effect of PGRN on mutant human APP in the young male mice.

2.) Were any sex differences in microglia observed in GrnKO mice that didn't carry the 5XFAD transgene?

Reply: So far we have not seen any sex differences of PGRN in microglia without 5XFAD transgene. We think the sex difference is largely caused by a male specific effect of PGRN on the mutant APP expressed from the transgene in neurons.

3.) Female GrnKO:5XFAD mice show striking changes in microglia, but no change in amyloid plaque load or behavior versus Grn WT:5XFAD mice. While there is extensive data showing the importance of microglia in AD pathogenesis, what is the data from this study to establish a link between microglial changes due to progranulin KO and amyloid pathology or behavioral deficits?

Reply: Thank you. These are excellent questions! The role of microglia in A-beta pathology is very complicated and still not well understood. Microglia can internalize A β but not necessarily degrade A β due to poor lysosomal acidification. Some studies have shown that microglial activities are important for the development of A-beta plaques, while other ones have shown that microglial activation regulates the conversion between diffuse vs dense core plaques. Although PGRN loss do not have a major effect on A β dynamics, it does seem to reduce the number and area of A β plaques without affecting overall A β levels at 10 month-old, indicating that A β is more concentrated at each plaque.

In response to A β , microglia also undergo activation, which can lead to the secretion of toxic cytokines and aberrant pruning of synapses. PGRN deficiency has been shown to result in increased microglial activation and synaptic pruning, which could have detrimental effects in AD mouse models. Since PGRN deficient male mice have reduced A-beta and enhanced microglial activation at the same time, it is very hard to predict outcomes in behavioral tests. Our results do show a trend of improved behavior in the Y-maze in young PGRN knockout males compared to age matched controls, but it didn't reach statistical significance.

March 26, 2021

Re: Life Science Alliance manuscript #LSA-2020-00874-TR-A

Dr. Fenghua Hu
Cornell University
Weill Institute for Cell and Molecular Biology and Department of Molecular Biology and Genetics
345 Weill Hall, Cornell University
Ithaca, NY 14853

Dear Dr. Hu,

Thank you for submitting your revised manuscript entitled "A multifaceted role of progranulin in regulating amyloid-beta dynamics and responses" to Life Science Alliance. The manuscript has been seen by the original reviewers whose comments are appended below. While the reviewers continue to be overall positive about the work in terms of its suitability for Life Science Alliance, some important issues remain.

We apologize for this delay in getting back to you. As you will note from the reviewers' comments below, while Reviewer 3 is satisfied with the revised manuscript, Reviewer 1 still has some concerns. Given that these concerns have arisen from the new data that has been included in the revised manuscript, we would like to give you the opportunity to address this in a final revision. The first 2 points raised by Reviewer 1 could be potentially addressed by a detailed discussion, pts 3-10 should be addressed with improved data.

Our general policy is that papers are considered through only one revision cycle; however, given that the suggested changes are relatively minor, we are open to one additional short round of revision. Please note that I will expect to make a final decision without additional reviewer input upon resubmission.

Please submit the final revision within one month, along with a letter that includes a point by point response to the remaining reviewer comments.

- A letter addressing the reviewers' comments point by point.
- An editable version of the final text (.DOC or .DOCX) is needed for copyediting (no PDFs).
- High-resolution figure, supplementary figure and video files uploaded as individual files: See our

detailed guidelines for preparing your production-ready images, <https://www.life-science-alliance.org/authors>

B. MANUSCRIPT ORGANIZATION AND FORMATTING:

Sincerely,

Shachi Bhatt, Ph.D.

Executive Editor

Life Science Alliance

<https://www.lsjournal.org/>

Interested in an editorial career? EMBO Solutions is hiring a Scientific Editor to join the international Life Science Alliance team. Find out more here -

https://www.embo.org/documents/jobs/Vacancy_Notice_Scientific_editor_LSA.pdf

Reviewer #1 (Comments to the Authors (Required)):

The manuscript by Du et al describing the effect of PGRN deletion in 5XFAD mice has been revised. The Editor has apparently deemed that issues related to lack of Novelty, Behavioral Effect and Mechanism of Sex Variation are not essential to publication in LSA, and these issues remain unaddressed. The editorial suggestion was to focus on quantification of A β pathology, and PGRN effects on microglia. The revised manuscript includes A β ELISA from 4 and 10 months as well as A β plaque measures at 10 months. There are new data showing a sex specific effect on APP levels with respect to PGRN genotype. Any effect of PGRN on microglial near plaques is now reported as sex-independent.

Unfortunately, the revisions seem to have made the study less, rather than more, clear.

1. A β measurement: New A β ELISA measurements were added in Fig. 2A (4 months) and 3B (10 months) and A β plaque histology measures at 10 months to Fig. 3A. These data are confusing and not consistent. At 10 months, the ELISA shows no difference between 5XFAD mice with or without Grn (regardless of sex). The A β plaque measures show a slight but significant reduction of area in both males and females. At early ages (4 mo) with low A β accumulation, the A β ELISA matches the previous histology. However, the accumulation of A β to levels adequate to impair brain function during aging does not seem to be substantially affected by PGRN.

2. APP Metabolism: The authors present new data that there is a selective reduction of human transgenic APP, but not mouse APP or secretases, and that this is specific for young male Grn knockouts. The authors suggest that Grn regulates "APP metabolism". The more obvious possibility is that the Thy1 transgene is differentially regulated by sex and/or PGRN. No assessment of mRNA levels are made here. A previous study concluded that there are estrogen-sensitive elements in the Thy1 promoter driving APP in the 5XFAD mice (Sadlier et al., Mol Neurodegeneration 10, 1 (2015)). If this is the case, and there is an interaction of PGRN with this transgene-specific anomaly, this obviously has no relevance for human AD.

There may in fact be no interaction of PGRN with sex in regard to A β accumulation. Instead, it may be that at very low levels (as in 4 month males), PGRN has a detectable effect, but at higher levels (as in 4 month females and 10 month of both sexes) there is no reliable effect of PGRN on A β accumulation. As it stands, the new "mechanistic" data substantially reduce the impact of any potential sex:PGRN interaction with regard to AD. This reduced impact is further supported by the absence of any PGRN effect on microglial to A β plaque added to Fig. 4F and 9B.

Importantly, the conclusion that a loss of PGRN alters APP metabolism in 5XFAD is not adequately supported to be justified. Studies of APP fragments, such as CTFs, sAPP α and sAPP β , as well as APP mRNA, would be required.

3. Immunoblot data presentation: There are several issues.

- In Figure S6, it looks like the blot panels were made by jointing two separate membranes or cropping some lanes. If that is the case, the authors must put a dividing line between them.
- In Figure 4 and 5, duplicated GAPDH panels were presented. If the experiments were performed simultaneously, it should be noted in the legend.
- Some of blots have multiple nonspecific spot-like signals overlapping with specific signals. This includes Fig 2B, Fig 4C, and Fig 7A. How were those blots quantified? Especially, the LAMP1 blot in Fig 7A seems too dirty to measure a specific signal.

4. In Fig 2B and C, the authors report ~50% reduction in human full-length (FL) APP levels in 5XFAD Grn $^{-/-}$ mice, but no change in total (human and mouse) FL APP levels. However, expression of the APP transgene in 5XFAD mice is known to be at least as great as endogenous APP in WT mice (Oakley et al J Neurosci, 2006). Thus, it seems inconsistent to find no change in total (human and mouse) FL APP levels.

5. Unfortunately, the A β plaque area measures are normalized, and are not reported as an absolute values, which is standard.

6. (Related to the previous #7) If a correlation was not observed between TFE3 nuclear signal and galactin-3 expression, representative images of galactin-3 in Figure 9D need to be removed. The images are not consistent with the description in pg.8, lines 2-4.

7. (Related to the previous #10) In Figure S2, the number of animals in the legend does not seem consistent with dots in the figure.

8. (Related to the previous #10) The basis for assuming normality (Gaussian distribution) was requested in the original #10. This was not about "normalization", and has not been addressed.

9. (Related to the previous #13) In Figure 9D, the label is still Dapi while the method says Hoechst.

10. There are inappropriate citations.

a) Pg.4, line 32: Mendasaikhan et al. have not investigated any of AD mouse models.

b) Pg.6, lines 17,18 and pg.11, lines 20-23: Huang et al. showed that galectin-3 and GPNMB are up-regulated in Grn^{-/-} mice and FTD-GRN. That study had nothing to do with AD. On the other hand, the present study has not found an increase of those proteins in Grn^{-/-} mice even at 10 months of age. Thus, their results are NOT consistent with the previous study reported by Huang et al.

Reviewer #3 (Comments to the Authors (Required)):

In this revised manuscript, Du et al have been very responsive to reviewer feedback, resulting in a significantly improved paper that will be a useful addition to the literature on progranulin and AD.

The two main findings of the paper are as follows:

- 1.) Progranulin deficiency induces a sex-specific difference in APP metabolism in the 5XFAD mouse model.
- 2.) Regardless of sex, progranulin deficiency dramatically changes the microglial response to amyloid beta. This is characterized by greater microglial activation and higher expression of multiple lysosomal proteins.

Both of these findings are well supported by the data.

My only remaining comment is that the authors should dedicate a paragraph or two in the discussion to the fact that the dramatic microglial changes in Grn KO 5XFAD mice do not seem to translate to changes in plaques, amyloid beta levels, or behavior deficits, as the sex differences in amyloid seem to be largely explained by APP metabolism. This seems very surprising given the extensive data on the role of microglia in AD pathogenesis and deficits in AD mouse models. The authors' data are very solid and support their conclusions, but it would be nice to see acknowledgement of this surprising result and some discussion of why the microglial changes seem to be dissociated from the amyloid changes in their mouse model.

Reviewer #1 (Comments to the Authors (Required)):

The manuscript by Du et al describing the effect of PGRN deletion in 5XFAD mice has been revised. The Editor has apparently deemed that issues related to lack of Novelty, Behavioral Effect and Mechanism of Sex Variation are not essential to publication in LSA, and these issues remain unaddressed. The editorial suggestion was to focus on quantification of A β pathology, and PGRN effects on microglia. The revised manuscript includes A β ELISA from 4 and 10 months as well as A β plaque measures at 10 months. There are new data showing a sex specific effect on APP levels with respect to PGRN genotype. Any effect of PGRN on microglial near plaques is now reported as sex-independent.

Unfortunately, the revisions seem to have made the study less, rather than more, clear.

1. A β measurement: New A β ELISA measurements were added in Fig. 2A (4 months) and 3B (10 months) and A β plaque histology measures at 10 months to Fig. 3A. These data are confusing and not consistent. At 10 months, the ELISA shows no difference between 5XFAD mice with or without Grn (regardless of sex). The A β plaque measures show a slight but significant reduction of area in both males and females. At early ages (4 mo) with low A β accumulation, the A β ELISA matches the previous histology. However, the accumulation of A β to levels adequate to impair brain function during aging does not seem to be substantially affected by PGRN.

Reply: Thanks for the nice summary of our findings. We agree that the age dependent regulation of A-beta by PGRN in the male mice can be a little confusing and PGRN does not seem to have a big effect in the 10 month-old mice with high A-beta levels.

2. APP Metabolism: The authors present new data that there is a selective reduction of human transgenic APP, but not mouse APP or secretases, and that this is specific for young male Grn knockouts. The authors suggest that Grn regulates "APP metabolism". The more obvious possibility is that the Thy1 transgene is differentially regulated by sex and/or PGRN. No assessment of mRNA levels are made here. A previous study concluded that there are estrogen-sensitive elements in the Thy1 promoter driving APP in the 5XFAD mice (Sadlier et al., Mol Neurodegeneration 10, 1 (2015)). If this is the case, and there is an interaction of PGRN with this transgene-specific anomaly, this obviously has no relevance for human AD.

There may in fact be no interaction of PGRN with sex in regard to A β accumulation. Instead, it may be that at very low levels (as in 4 month males), PGRN has a detectable effect, but at higher levels (as in 4 month females and 10 month of both sexes) there is no reliable effect of PGRN on A β accumulation. As it stands, the new "mechanistic" data substantially reduce the impact of any potential sex:PGRN interaction with regard to AD. This reduced impact is further supported by the absence of any PGRN effect on microglial to A β plaque added to Fig. 4F and 9B.

Importantly, the conclusion that a loss of PGRN alters APP metabolism in 5XFAD is not adequately supported to be justified. Studies of APP fragments, such as CTFs, sAPPalpha and sAPPbeta, as well as APP mRNA, would be required.

Reply: Thanks for the suggestions. Since the levels of PS1 is not changed in the male Grn-/- mice and the PS1 expression is also driven by the Thy1 promoter, we'd like to think that the regulation of APP levels by PGRN is not due to difference in Thy1 promoter activities.

We agree that the microglial responses to A-beta do not seem be sex-dependent in the absence of PGRN. But our data strongly argue that there is a sex dependent regulation of mutant human APP levels by PGRN in the young male mice.

3. Immunoblot data presentation: There are several issues.

- In Figure S6, it looks like the blot panels were made by jointing two separate membranes or cropping some lanes. If that is the case, the authors must put a dividing line between them.
- In Figure 4 and 5, duplicated GAPDH panels were presented. If the experiments were performed simultaneously, it should be noted in the legend.
- Some of blots have multiple nonspecific spot-like signals overlapping with specific signals. This includes Fig 2B, Fig 4C, and Fig 7A. How were those blots quantified? Especially, the LAMP1 blot in Fig 7A seems too dirty to measure a specific signal.

Reply: In Figure S6, these samples were in one membrane but the order was reorganized to match with the order in other figure panels. According to the reviewer's suggestion, we have put a dividing line between them.

In Figure 4 and 5, the experiments were performed simultaneously, we have explained it in the legend.

The blots have been quantified by subtracting signals from a region with similar background. Most of the blots we have used for quantification do not have the non-specific spot like signals and we have selected more representative western blot images for Fig 2B, Fig 4C, and Fig 7A.

4. In Fig 2B and C, the authors report ~50% reduction in human full-length (FL) APP levels in 5XFAD Grn^{-/-} mice, but no change in total (human and mouse) FL APP levels. However, expression of the APP transgene in 5XFAD mice is known to be at least as great as endogenous APP in WT mice (Oakley et al J Neurosci, 2006). Thus, it seems inconsistent to find no change in total (human and mouse) FL APP levels.

Reply: The antibody we have used to detect mouse APP (Proteintech Group, 25524-1-AP) seems to have higher affinity for mouse APP than in human APP. Or the human APP is expressed at relatively low level in the 4.5-month-old mice compared to the endogenous mouse APP. The total APP levels with or without the human APP transgene do not seem to differ from each other in the 4.5-month-old mice in the Western blot with the Proteintech antibody (Figure S3B). Thus we have changed the label m+h to m APP in figure 2 to avoid confusions.

5. Unfortunately, the A β plaque area measures are normalized, and are not reported as an absolute values, which is standard.

Reply: We understand the reviewer's concerns. However, to be consistent with the quantification of A β plaque number and intensity, which were normalized to age-matched 5XFAD controls, and to save space in the figure, we've decided to only show the normalized value of the plaque area.

6. (Related to the previous #7) If a correlation was not observed between TFE3 nuclear signal and galactin-3 expression, representative images of galactin-3 in Figure 9D need to be removed. The images are not consistent with the description in pg.8, lines 2-4.

Reply: We have removed galectin-3 channel in Fig. 9D and made edits in the text accordingly.

7. (Related to the previous #10) In Figure S2, the number of animals in the legend does not seem consistent with dots in the figure.

Reply: Thanks for noticing that. We have fixed the mistake.

8. (Related to the previous #10) The basis for assuming normality (Gaussian distribution) was requested in the original #10. This was not about "normalization", and has not been addressed.

Reply: We have added "The normality of data was tested by the Shapiro–Wilk test ($P>0.05$)" in the statistical analysis method session.

9. (Related to the previous #13) In Figure 9D, the label is still Dapi while the method says Hoechst.

Reply: Thanks for noticing that. We have fixed the mistake.

10. There are inappropriate citations.

a) Pg.4, line 32: Mendasaikhan et al. have not investigated any of AD mouse models.

b) Pg.6, lines 17,18 and pg.11, lines 20-23: Huang et al. showed that galectin-3 and GPNMB are up-regulated in Grn^{-/-} mice and FTD-GRN. That study had nothing to do with AD. On the other hand, the present study has not found an increase of those proteins in Grn^{-/-} mice even at 10 months of age. Thus, their results are NOT consistent with the previous study reported by Huang et al.

Reply: Thanks for your careful review. You are right that Mendasaikhan et al. have not investigated any of AD mouse models but they have shown PGRN expression in microglia surrounding A-beta plaques in human patient samples.

Regarding galectin-3 and GPNMB, you are correct that we have not found changes in these proteins in Grn^{-/-} mice without APP and PS1 transgene expression. However, in the 5XFAD mice, these two proteins show most upregulation, which is consistent with the data reported by Huang et al. in Grn^{-/-} mice and FTLD-GRN patients, showing that galectin-3 and GPNMB are the most upregulated proteins in response to PGRN loss.

Reviewer #3 (Comments to the Authors (Required)):

In this revised manuscript, Du et al have been very responsive to reviewer feedback, resulting in a significantly improved paper that will be a useful addition to the literature on progranulin and AD. The two main findings of the paper are as follows:

- 1.) Progranulin deficiency induces a sex-specific difference in APP metabolism in the 5XFAD mouse model.
- 2.) Regardless of sex, progranulin deficiency dramatically changes the microglial response to amyloid beta. This is characterized by greater microglial activation and higher expression of multiple lysosomal proteins.

Both of these findings are well supported by the data.

My only remaining comment is that the authors should dedicate a paragraph or two in the discussion to the fact that the dramatic microglial changes in Grn KO 5XFAD mice do not seem to translate to changes in plaques, amyloid beta levels, or behavior deficits, as the sex differences in amyloid seem to be largely explained by APP metabolism. This seems very surprising given the extensive data on the role of microglia in AD pathogenesis and deficits in AD mouse models. The authors' data are very solid and support their conclusions, but it would be nice to see acknowledgement of this surprising result and some discussion of why the microglial changes seem to be dissociated from the amyloid changes in their mouse model.

Reply: Thanks for your kind comments and suggestion. We have modified the discussion accordingly.

May 20, 2021

RE: Life Science Alliance Manuscript #LSA-2020-00874-TRR

Dr. Fenghua Hu
Cornell University
Weill Institute for Cell and Molecular Biology and Department of Molecular Biology and Genetics
345 Weill Hall, Cornell University
Ithaca, NY 14853

Dear Dr. Hu,

Thank you for submitting your revised manuscript entitled "A multifaceted role of progranulin in regulating amyloid-beta dynamics and responses". We would be happy to publish your paper in Life Science Alliance pending final revisions necessary to meet our formatting guidelines.

We apologize for this delay in getting back to you. We appreciate the changes and clarifications you have provided in response to the concerns from Reviewer 1. We would encourage you to include the points made and caveats mentioned in response to Rev 1's pts 1 and 2 in the final revision of the manuscript.

Along with the points mentioned above, please also attend to the following:

- please use the [10 author names, et al.] format in your references (i.e. limit the author names to the first 10)
- please add your supplementary figure legends to the main manuscript text after the main figure legends
- please be sure that all Authors are mentioned in Author's contribution section in the manuscript text
- please add a conflict of interest statement to your main manuscript text
- please upload your supplementary figures as single files as well
- we encourage you to revise the figure legends for Supplementary figures such that the figure panels are introduced in an alphabetical order
- please add callouts for Figures 7A and B; 11A and B; S6A and B to your main manuscript text
- please provide better quality blots for Figure 2B
- please provide the original uncropped gels (source data) for Galectin-3 blots shown in Figure 4C
- please deposit the RNA seq data in an open data repository and provide the accession information under "data availability" section in the manuscript (<https://www.life-science-alliance.org/manuscript-prep#datadepot>)

A. FINAL FILES:

B. MANUSCRIPT ORGANIZATION AND FORMATTING:

Thank you for this interesting contribution, we look forward to publishing your paper in Life Science

Alliance.

Sincerely,

Shachi Bhatt, Ph.D.

Executive Editor

Life Science Alliance

<http://www.lsjournal.org>

May 26, 2021

RE: Life Science Alliance Manuscript #LSA-2020-00874-TRRR

Dr. Fenghua Hu
Cornell University
Weill Institute for Cell and Molecular Biology and Department of Molecular Biology and Genetics
345 Weill Hall, Cornell University
Ithaca, NY 14853

Dear Dr. Hu,

Thank you for submitting your Research Article entitled "A multifaceted role of progranulin in regulating amyloid-beta dynamics and responses". It is a pleasure to let you know that your manuscript is now accepted for publication in Life Science Alliance. Congratulations on this interesting work.

DISTRIBUTION OF MATERIALS:

Again, congratulations on a very nice paper. I hope you found the review process to be constructive and are pleased with how the manuscript was handled editorially. We look forward to future exciting submissions from your lab.

Sincerely,

Shachi Bhatt, Ph.D.

Executive Editor

Life Science Alliance

<http://www.lsjournal.org>
